



# The challenge of simulating the sensitivity of the Amazonian clouds microstructure to cloud condensation nuclei number concentrations

Pascal Polonik[1,*], Christoph Knote[1], Tobias Zinner[1], Florian Ewald[2], Tobias Kölling[1], Bernhard Mayer[1], Meinrat O. Andreae[3,4], Tina Jurkat-Witschas[4], Thomas Klimach[4], Christoph Mahnke[5,6], Sergej Molleker[5], Christopher Pöhlker[4], Mira L. Pöhlker[4], Ulrich Pöschl[4], Daniel Rosenfeld[7], Christiane Voigt[2,6], Ralf Weigel[6], and Manfred Wendisch[8]

[1]Meteorologisches Institut, Ludwig Maximilians-Universität München, Munich, Germany
[2]Institut für Physik der Atmosphäre, Deutsches Zentrum für Luft- und Raumfahrt (DLR), Oberpfaffenhofen, Germany
[3]Scripps Institution of Oceanography, University of California at San Diego, La Jolla, California, USA
[4]Multiphase Chemistry and Biogeochemistry Departments, Max Planck Institute for Chemistry, Mainz, Germany
[5]Particle Chemistry Department, Max Planck Institute for Chemistry, Mainz, Germany
[6]Institut für Physik der Atmosphäre, Johannes Gutenberg-Universität, Mainz, Germany
[7]Institute of Earth Sciences, Hebrew University of Jerusalem, Jerusalem, Israel
[8]Leipziger Institut für Meteorologie, Universität Leipzig, Leipzig, Germany
[*]*now at:* Scripps Institution of Oceanography, University of California at San Diego, La Jolla, California, USA

**Correspondence:** Christoph Knote (christoph.knote@physik.uni-muenchen.de)

**Abstract.** The realistic representation of cloud-aerosol interactions is of primary importance for accurate climate model projections. The investigation of these interactions in strongly contrasting clean and polluted atmospheric conditions in the Amazon area has been one of the motivations for several field observations, including the airborne Aerosol, Cloud, Precipitation, and Radiation Interactions and DynamIcs of CONvective cloud systems - Cloud Processes of the Main Precipitation Systems in Brazil:

A Contribution to Cloud Resolving Modeling and to the GPM (Global Precipitation Measurement) (ACRIDICON-CHUVA) campaign based in Manaus, Brazil in September 2014. In this work we combine in situ and remotely sensed aerosol, cloud, and atmospheric radiation data collected during ACRIDICON-CHUVA with regional, online-coupled chemistry-transport simulations to evaluate the model's ability to represent the indirect effects of biomass burning aerosol on cloud microphysical properties (droplet number concentration and effective radius).

We found agreement between modeled and observed median cloud droplet number concentrations (CDNC) for low values of CDNC, i.e., low levels of pollution. In general, a linear relationship between modeled and observed CDNC with a slope of two was found, which means a systematic underestimation of modeled CDNC as compared to measurements. Variability in cloud condensation nuclei (CCN) number concentrations and cloud droplet effective radii ($r_{\mathrm{eff}}$) was also underestimated by the model.

Modeled effective radius profiles began to saturate around 500 CCN $\mathrm{per\,cm}^3$ at cloud base, indicating an upper limit for the model sensitivity well below CCN concentrations reached during the burning season in the Amazon Basin. Regional background aerosol concentrations were sufficiently high such that the additional CCN emitted from local fires did not cause a notable change in modelled cloud microphysical properties.





In addition, we evaluate a parameterization of CDNC at cloud base using more readily available cloud microphysical properties, aimed at in situ observations and satellite retrievals. Our study casts doubt on the validity of regional scale modeling studies of the cloud albedo effect in convective situations for polluted situations where the number concentration of CCN is greater than $500\ \mathrm{cm}^{-3}$.

## 1 Introduction

Aerosol particles influence the formation of cloud droplets, and thereby the microphysical and macrophysical properties of clouds. Cloud droplet sizes and number concentrations determine the effect of clouds on atmospheric radiation and, therefore, also on weather and climate. Increased aerosol concentrations increase the cloud albedo (Twomey, 1991) and possibly the lifetime (Albrecht, 1989) of clouds by decreasing droplet size if the total liquid water mass is assumed constant. These indirect effects lead to increased cloud albedo and, thus, enhanced reflection of solar radiation under high aerosol loading, and therefore causes a net cooling of the sub-cloud layer. However, the magnitude of these effects is not well constrained, which causes major uncertainties in current climate projections (IPCC, 2014).

Representing aerosol-cloud interactions in numerical models that form the basis of these projections is challenging because two of the most dynamic and complex atmospheric systems (aerosol and clouds) must be adequately represented individually before considering an accurate representation of their interactions. Multiple processes are involved, such as activation of aerosol particles to cloud droplets, phase-transfer, evaporation, and wet deposition. Parameterizations with varying levels of complexity have been incorporated into numerical models of the atmosphere (e.g., Thompson et al., 2008; Morrison et al., 2005), which has led to improved short-term forecasts in certain case studies (Zhang et al., 2010). It is difficult, however, to disentangle benefits in forecast-relevant quantities (e.g., $500\ \mathrm{hPa}$ pressure field deviation or storm track accuracy) from an actual improvement in the modelled cloud macro- and microphysical characteristics and its impact on the atmospheric radiation budget. Testing such parameterizations on a mechanistic level requires direct comparisons of model output to a variety of data sources (Seinfeld et al., 2016) as well as situations in which a noticeable aerosol signal can be expected. Events like volcanic eruptions (Malavelle et al., 2017; McCoy and Hartmann, 2015), desert dust outbreaks (Levin et al., 2005; Sassen et al., 2003), or wildfires (Rosenfeld, 1999; Brioude et al., 2009) provide strong signals that facilitate such process-level analysis of aerosol-cloud interactions.

In this work we present a case study that uses simulations and novel measurements of a recent field campaign to explore aerosol-cloud-radiation interactions during the biomass burning season in the Amazon region. The Aerosol, Cloud, Precipitation, and Radiation Interactions and DynamIcs of CONvective cloud systems - Cloud Processes of the Main Precipitation Systems in Brazil: A Contribution to Cloud Resolving Modeling and to the GPM (Global Precipitation Measurement) (ACRIDICON-CHUVA) field campaign (Wendisch et al., 2016) was conducted over the Amazon in September 2014 during the dry season, when biomass burning from regional agricultural practices creates strong perturbations of cloud condensation





nuclei (CCN) number concentration (Pöhlker et al., 2018). We use remote sensing and in situ data collected by the High Al-
titude and Long Range Research Aircraft (HALO), operated by the German Aerospace Center (DLR) to evaluate our model
predictions. Amongst other measurements, aerosol size and composition, CCN concentration, cloud phase and droplet size, and
trace gas concentrations were collected. HALO flew underneath and within clouds to reconstruct vertical profiles. Typically,
HALO research flights began with a ferry from Manaus to a region of interest and then back to Manaus (Figure 1, Table 1).

Regions of interest were areas with forecasted presence of convective clouds above specific surface conditions, such as intact
forest or polluted agricultural burning areas. Many of the HALO flights were conducted in regions where medium or high
aerosol number concentrations from biomass burning were suspected to influence cloud microphysical and radiative properties
(Table 1).

We tried to reproduce the measurements conducted during the HALO flights by numerical simulations using the Weather

Research and Forecasting model with Chemistry (WRF-Chem, Grell et al., 2005) at convection-permitting scales. The sim-
ulations feature a size-resolved description of the full lifecycle of ambient aerosol, including biomass burning emissions,
secondary particle formation through trace gas oxidation, and dry and wet deposition. Radiative properties of the aerosol popu-
lation are considered based on size distribution and component-resolved optical properties (Barnard et al., 2010). The modeled
aerosol description is linked to the detailed cloud microphysics parameterization of Morrison and Gettelman (2008). The num-

ber of CCN available for cloud formation as well as their physicochemical properties (size distribution and hygroscopicity) are
provided to the cloud microphysics scheme based on the online-calculated aerosol properties. Conversely, activation of aerosol
particles to cloud droplets leads to their removal from the aerosol-phase. Transport within cloud droplets, aqueous-phase chem-
istry, and washout by rain is explicitly represented in the model.

We first evaluate whether numerical simulations on convection permitting scales can accurately represent the observed

cloud microphysical properties. For this purpose we focus on cloud droplet number concentration (CDNC) and cloud droplet
effective radius ($r_{\mathrm{eff}}$) vertical profiles. $r_{\mathrm{eff}}$ profiles are representative of the microphysical development of a cloud and can be
derived from in situ as well as remote sensing observations. Here we use in situ measurements of droplet size and number
concentration along HALO flight tracks rearranged into profiles, and retrievals of $r_{\mathrm{eff}}$ profiles from passive remote sensing
observations (Ewald et al., 2018).

While $r_{\mathrm{eff}}$ profiles describe the vertical evolution of cloud microphysical properties, it is actually the number of activated
cloud condensation nuclei at cloud base, $N_{\mathrm{a}}$, that provides the link between cloud development and aerosol availability. As
$N_{\mathrm{a}}$ is a somewhat elusive quantity to observe, especially from satellites, parameterizations have been suggested to determine
$N_{\mathrm{a}}$ based on observations of $r_{\mathrm{eff}}$ higher up in a cloud (Rosenfeld et al., 2012). In the second part of this work we evaluate the
applicability of the parameterization from Freud et al. (2011) using in situ, remote-sensing and model-derived $r_{\mathrm{eff}}$ profiles.



**Table 1.** Dates of flights conducted during the ACRIDICON-CHUVA campaign, with basic information about each flight compiled from Wendisch et al. (2016) and the campaign blog (https://acridicon-chuva.weebly.com/; last accessed: July 10, 2018). CCN levels during each research flight are binned into low ("+"), medium ("++") and high ("+++").

| Date | Flight # | CCN level | Description |
|------|----------|-----------|-------------|
| 2014-09-11 | AC09 | + | Clean conditions for cloud profiling |
| 2014-09-12 | AC10 | + | Satellite coordination and several in situ clouds sampled in relatively clean conditions |
| 2014-09-16 | AC11 | ++ | Tracer experiment near Manaus, with some fires in the vicinity |
| 2014-09-18 | AC12 | +++ | Polluted conditions but relatively few large clouds sampled |
| 2014-09-19 | AC13 | +++ | Polluted conditions, sampling of complete cloud profiles |
| 2014-09-21 | AC14 | ++ | Satellite coordination, GoAmazon GI aircraft coordination, medium pollution |
| 2014-09-23 | AC15 | ++ | Surface albedo measurement early, cloud sampled later, medium pollution |
| 2014-09-25 | AC16 | ++ | Tracer experiment near Manaus, fires in the vicinity |
| 2014-09-27 | AC17 | +++ | Sample clouds over different land surfaces, compare to GPM satellite, polluted conditions |
| 2014-09-28 | AC18 | + | Medium sized cumulus samples and full cloud profiles in clean conditions |

## 2 Methods

### 2.1 Model

The Weather Research and Forecast model with Chemistry (WRF-Chem, Grell et al., 2005) was used to simulate atmospheric motion while incorporating online calculations of trace gases and aerosol physical and chemical processes in a nested domain setup. We used the Model for OZone And Related chemical Tracers (MOZART) gas-phase chemistry (Emmons et al., 2010; Knote et al., 2014) and the Model for Simulating Aerosol Interactions and Chemistry (MOSAIC) aerosol module (Zaveri et al., 2008), with a volatility basis set parameterization for organic aerosol evolution (Knote et al., 2015). Anthropogenic emissions data were taken from the Emissions Database for Global Atmospheric-Research from the task force for Hemispheric Transport of Air Pollution (EDGAR-HTAP, Janssens-Maenhout et al., 2012). Biogenic emissions are calculated online using the Model of Emissions of Gases and Aerosols from Nature (MEGAN, Guenther et al., 2006). The Fire Inventory from NCAR (FINN) module was used for the fire emissions data (Wiedinmyer et al., 2011). One degree resolution, six-hourly updated meteorological boundary conditions were taken from analyses of the National Center For Environmental Prediction Global Forecast System (NCEP GFS), and chemical boundary conditions were provided by forecasts of the global chemistry model MOZART (https://www.acom.ucar.edu/wrf-chem/mozart.shtml, last accessed February 6th, 2018). Cloud microphysical properties were represented by the double-moment microphysics scheme by Morrison et al. (2009), and no convection parameterization was applied in the nested domain. The Morrison scheme has five hydrometeor classes (cloud droplets, rain, cloud ice, snow, and



graupel), with each size distribution parameterized by a Gamma function. The cloud droplet effective radius is calculated through integration over the droplet size distribution:

$$r_{eff} = \frac{\int_0^\infty r^3 N(r)\mathrm{d}r}{\int_0^\infty r^2 N(r)\mathrm{d}r} \tag{1}$$

with $r$ cloud droplet radius, and $N(r)$ droplet number concentration at radius $r$.

Effects of aerosol particles on atmospheric radiation (direct effect) are considered as presented in Fast et al. (2006). Activation of aerosol particles as cloud droplets is calculated based on the aerosol size distribution and chemical composition using $\kappa$-Koehler theory (Abdul-Razzak and Ghan, 2000, 2002), with relevant aspects of the implementation in the version of WRF-Chem used here presented in Gustafson Jr et al. (2007) and Chapman et al. (2009). The life cycle of activated aerosol particles is modelled explicitly; i.e., they are removed from the interstitial aerosol population and their evolution is modelled
in accordance with that of the cloud droplets in which they are incorporated in, including processes like washout from precipitation or re-evaporation. Cloud chemistry and limited heterogeneous processes are included as presented in Knote et al. (2015). Chemistry and aerosol processes are included in an operator-splitting fashion, in which individual processes update model fields sequentially. In each WRF-Chem time step, first advection is calculated, followed by droplet activation and finally the remaining chemistry and aerosol processes.

WRF-Chem simulations over the Amazon region were conducted for the ACRIDICON-CHUVA mission period between 8 - 30 September 2014. A continuous simulation with 15 km horizontal resolution, covering an area of approximately 3000 $\times$ 2700 km$^2$ (200 $\times$ 180 grid points), and 36 vertical levels up to 50 hPa, was conducted for the full campaign period (see Figure 1 for domain overview). To keep the large-scale meteorology in line with reality, WRF-Chem was restarted every 24 hours (at 0 hours UTC) from GFS analyses. Concentrations of trace gases and aerosol quantities were carried over, however,
to allow for multi-day pollution build-up and aging. Each 24 hour period was simulated with a 6 hour meteorological spin-up with nudging and a chemical restart file from the previous day. Meteorology was then allowed to evolve freely within the WRF-Chem domain (i.e. no nudging was applied) to enable the model to develop the implemented aerosol-cloud-interactions. Three additional days before the study period were simulated to spin-up trace gas chemistry and aerosol.

  Convection-permitting, 3 km horizontal resolution domains (180 $\times$ 180 grid points, approx. 540 $\times$ 540 km$^2$) were then
"nested" into this simulation during days with HALO flights. Two-way interactions were allowed between the parent and the nested domains. The location of these "nests" varied and were chosen so that they covered the area of interest sampled by HALO in each flight (Figure 1, see also Section 3.1). On each flight day, the nested domain was started (by interpolating the current state of the outer domain) at 09:00 UTC and run until 21:00 UTC, hence covering the full time frame of each HALO research flight. All model results presented in this study are from the nested, convection-permitting domains.





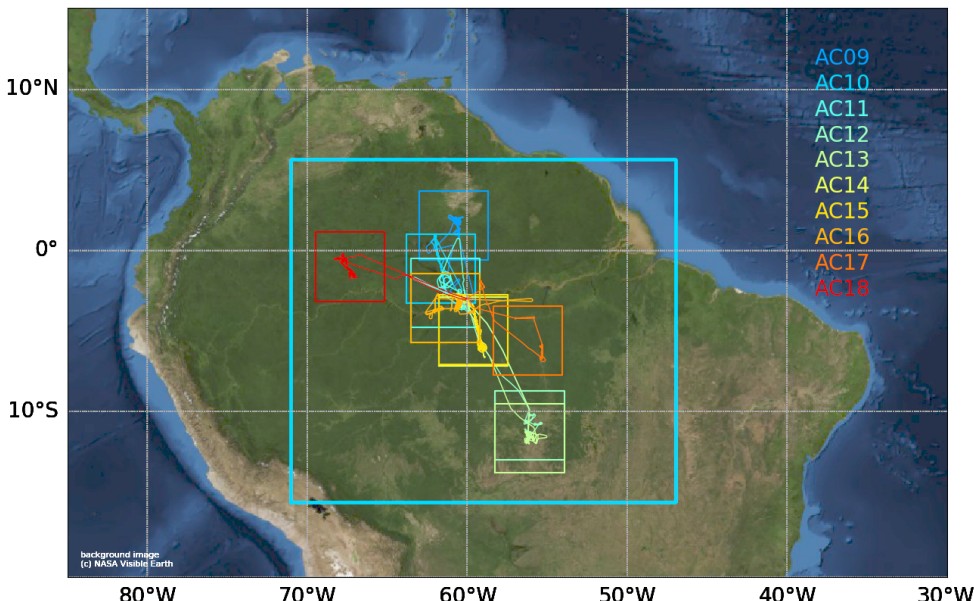

**Figure 1.** A map showing the campaign area, with all ACRIDICON-CHUVA research flights considered in this study as color-coded lines, the continuously-run outer simulation domain (blue box) as well as the individual nested domains used for analysis of each research flight, identified by the flight labels (Table 1).

## 2.2 Measurements

### 2.2.1 Cloud in situ measurements

The cloud combination probe (CCP) combines the cloud imaging probe (CIP) and the cloud droplet probe (CDP) to measure the cloud particle size distribution by detecting their forward-scattered laser light (Lance et al., 2010). During the ACRIDICON-CHUVA campaign, the CCP measured at 1 Hz frequency from underneath the right wing of the HALO aircraft (Wendisch et al., 2016). A correction for the high flight velocities was applied to improve data quality (Weigel et al., 2016). The CCP measures particles with diameters between 2 - 960 μm, but here we only used the 14 bins for particle diameters from 3 - 50 μm (from the CDP) to calculate the cloud particle effective radius. Except for the details of the selection of appropriate data points, the data used here is the same as described in Braga et al. (2017a). To filter the data we calculated liquid water content from binned effective diameter measurements and only included those with at least $1 \, \mathrm{g \, kg^{-1}}$ liquid water content. This threshold is consistent with the one used to define "cloudy" points in model output.

Like the CCP-CDP, the Cloud and Aerosol Spectrometer with Depolarization (CAS-DPOL) measures cloud particle size distributions at 1 Hz frequency (Baumgardner et al., 2011; Voigt et al., 2017). The CAS-DPOL measures the intensity of





forward-scattered light between 4 - 12 degrees in 30 size bins from particles with diameter 0.5 - 50 μm. The polarized backward-scattered light is used to analyze the sphericity and thermodynamic phase of the measured particles (Baumgard-
ner et al., 2014; Järvinen et al., 2016), but this capability was not used for our analysis. Our calculation of the cloud particle effective radius (Schumann et al., 2017) was again limited to particles between 3 - 50 μm, which corresponds to 10 Mie-ambiguity corrected size bins, to account for consistency with the CDP. Further details on CAS-DPOL data evaluation are given in Kleine et al. (2018).

Profiles of $r_{\mathrm{eff}}$ were derived using data from both the CAS-DPOL and the CDP. Braga et al. (2017a) demonstrated that
the CDP and CAS-DPOL instruments are comparable within their expected measurement uncertainties. Flamant et al. (2018) and Taylor et al. (2019) also found good agreement between CAS-DPOL and CDP measurements in shallow clouds. Here, we combine measurements from both instruments into one in situ dataset to construct effective radii profiles. Therefore, the concentration of activated cloud condensation nuclei $N_{\mathrm{a}}$, is derived using all in situ $r_{\mathrm{eff}}$ measurements with their respective adiabatic liquid water content (see further description in Section 2.2.4). Treating in situ measurements from the two instruments
as independent is justifiable in part because they are located on opposite wings of the aircraft.

### 2.2.2   CCN in situ measurements

The number concentration of CCN was measured with a continuous-flow streamwise thermal gradient CCN counter (CCNC, model CCN-200, DMT, Longmont, CO, USA) (Roberts and Nenes, 2005; Rose et al., 2008). Activated CCN that grow to a diameter of at least 1 μm at a set water vapor supersaturation between 0.1 - 5% are counted by the instrument at 1Hz. Two
sample inlets were used during the ACRIDICON-CHUVA campaign, but here we only use data from the HALO aerosol sub-micron inlet (HASI), which collected data at a constant supersaturation of 0.55 %. The uncertainty of the CCN measurements is dominated by the counting statistics and ranges between 10% for high CCNs and 20% for low CCNs (Krüger et al., 2014). The supersaturation uncertainty is also about 10% (Braga et al., 2017a).

### 2.2.3   Cloud remote sensing measurements

The spectral imager of the Munich Aerosol and Cloud Scanner (specMACS) was installed on the HALO aircraft during ACRIDICON-CHUVA. specMACS is a hyperspectral line camera that measures at visible and near-infrared wavelengths (Ewald et al., 2016). Marshak et al. (2006) and Martins et al. (2011) suggested using the solar radiation reflected by illuminated cloud sides to derive the vertical profile of effective radius and cloud phase, but the ACRIDICON-CHUVA campaign was the first time that passive cloud side remote sensing was applied systematically for a large number of cases. Zinner et al.
(2008) and Ewald et al. (2018) developed a cloud side retrieval and demonstrated the application using ACRIDICON-CHUVA data. Jäkel et al. (2017) derived phase information from cloud-side reflectivity measurements during ACRIDICON-CHUVA. specMACS was mounted on HALO at a sideward viewing port to observe clouds passed by the aircraft. Cloud vertical profiles were then retrieved using the method by Ewald et al. (2018) along the flight route akin to a push-broom satellite instrument. Results for three cases are compared to in situ and WRF-Chem model data.





specMACS cases shown in this paper are first example cases and mainly presented to showcase the capability of airborne remote sensing to provide effective radius profiles and cloud droplet number concentration (CDNC). They are not representative for whole flights or flight regions as the used in situ or modelled data, but show specific example local situations along a few minutes of flight time. In this respect they complement the large scale picture provided by modelled data averaged over $540 \times 540$ km$^2$ or the in situ data collected over several hours flight time. specMACS cloud scenes were selected based on
favorable data collection conditions. This includes minimal turning of the aircraft, favorable sunlight conditions, and high cloud coverage.

### 2.2.4   Derivation of $N_\mathrm{a}$ from in situ, remote sensing, and model cloud data

The central quantity to determine the influence of aerosol on cloud development and lifetime is the number of activated cloud condensation nuclei at cloud base, $N_\mathrm{a}$. During ACRIDICON-CHUVA, HALO directly sampled $N_\mathrm{a}$ during their cloud profile
flights, providing a valuable comparison. As the collection of in situ data is expensive and spatial coverage is limited, Rosenfeld et al. (2012) suggested to infer $N_\mathrm{a}$ at cloud base using other more readily available observations like satellite retrievals. Freud et al. (2011) proposed a parametrization that derives $N_\mathrm{a}$ from the vertical profile of droplet radii. To do this, an adiabatic liquid water content ($LWC_\mathrm{a}$) is calculated from cloud base pressure and temperature under the assumption that all water vapor above the saturation vapor pressure is condensed during the moist adiabatic ascent of a parcel at a fixed $N_\mathrm{a}$ can be derived using an
empirical relation between $r_\mathrm{eff}$ and the volumetric radius, $r_\mathrm{v}$ (i.e., $r_\mathrm{v} = 1.08 \cdot r_\mathrm{eff}$ as in Freud et al. (2011)), $LWC_\mathrm{a}$, and the density of water $\rho_\mathrm{w}$:

$$N_a = \frac{1}{\rho_w} \cdot \frac{3}{4\pi} \cdot \frac{LWC_a}{r_\mathrm{v}^3} \cdot 0.7 \tag{2}$$

The ratio of $LWC_\mathrm{a}$ and $r_\mathrm{v}^3$ is found as the slope of a linear regression through all available point pairs of $LWC_\mathrm{a}$ and $r_\mathrm{v}^3$ in the droplet size profile, forced through the origin. An additional mixing factor of 0.7 accounts for the imperfection of the adiabatic
assumption (Freud et al., 2011; Braga et al., 2017a). Freud et al. (2011) empirically derived this factor using in situ effective radius and LWC data from multiple previous field campaigns, including one in the Amazon. Although there was geographic diversity in the data used for the derivation, only one estimation was made which may introduce an unknown error in our studies. This could be especially relevant for remotely sensed data that measure cloud sides rather than a cloud cross-section. Nonetheless, we apply the same derivation and same mixing factor to all three available $r_\mathrm{eff}$ datasets: remotely sensed, in
situ, and model output. Applying this method to multiple data sources provides insights into the validity of this concept. The resulting $N_\mathrm{a}$ can also be used for direct comparison of the different input $r_\mathrm{eff}$ profiles.



## 3 Representation of cloud microphysics in the model

### 3.1 Deriving comparable quantities for model-measurement evaluation

Comparing the three different sources of information on cloud microphysical properties (model, remote-sensing, and in situ observations) is not straightforward. Colocating in situ and remote-sensing observations required observing a cloud using the side-facing specMACS, and then flying into this cloud to obtain respective in situ measurements. During ACRIDICON-CHUVA, cloud clusters had been identified for each research flight, which were then passed several times to allow for remote-sensing observations before probing these clusters in situ. This precludes direct comparison of individual clouds without diligent data selection, but allows for a statistical comparison of in situ data collected near the cluster and the corresponding remote-sensing observations. Simulations will not reproduce an individual (observed) cloud, but they will create a comparable, realistic regional environment with comparable clouds. Hence, the nested domains were chosen such that they are center on the cloud cluster chosen as target for an ACRIDICON-CHUVA research flight. Assuming a homogeneous environment within the model domain, a statistical comparison of all modelled clouds in the model domain with observations taken of the cloud cluster within the domain is reasonable. Therefore, we used all clouds within the respective nested model domain to derive model statistics. Observation statistics are based on all data collected within the spatial domain of the model nest. As mentioned above, statistics pertaining to in-cloud variables are restricted to data points with a liquid water content of more than $1\,\mathrm{g\,kg^{-1}}$ in both model and observations.

### 3.2 Cloud droplet number concentrations

Figure 2 shows median in situ measurements of CDNC during flights and the median CDNC values from the entire nested model domain corresponding to the flight. Modeled and measured CDNC match for lower values of $200\,\mathrm{cm^{-3}}$ (AC09), but diverge for higher values. There is a linear relationship between WRF-Chem results and observations, albeit below the one-to-one line, leading to a factor of two of underestimation of CDNC for the most polluted case investigated (AC12 with about 750 $\mathrm{cm^{-3}}$ observed).

### 3.3 Variability in modeled $r_{\mathrm{eff}}$ profiles

All WRF-Chem modeled $r_{\mathrm{eff}}$ data from the ten nested domains was combined and binned by cloud-base CCN concentration (Figure 3). Cloud-base CCN is defined as the modeled CCN concentration at 0.5 % supersaturation directly below the lowest cloudy pixel in a model column.

The binning of $r_{\mathrm{eff}}$ profiles shows that the modeled profiles correspond to theoretical expectations; clouds with more available CCNs have a $r_{\mathrm{eff}}$ profile that is shifted towards smaller values relative to those with less available CCNs. The response to CCN concentration saturates in the model around 500-600 $\mathrm{cm^{-3}}$, indicating that biomass burning effects will be nonlinear and strongest in relatively clean conditions. We did not find such a saturation effect for CDNC (Figure 2). Between 2 - 4 km above sea level, where the most model clouds occur, the slope of the profile also scales with available CCNs. The radius grows




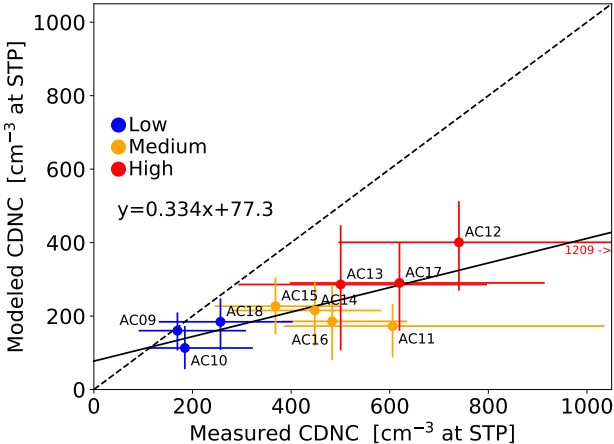

**Figure 2.** Median cloud droplet number concentration from the WRF domain and in situ measurements. The colors correspond to the CCN level labels in Table 1. Error bars depict the interquartile range (25 - 75% of all values). The equation describes the (solid black) regression line. The dashed black line is a 1-to-1 line for reference. STP refers to standard temperature (273.15K) and pressure (1000hPa).

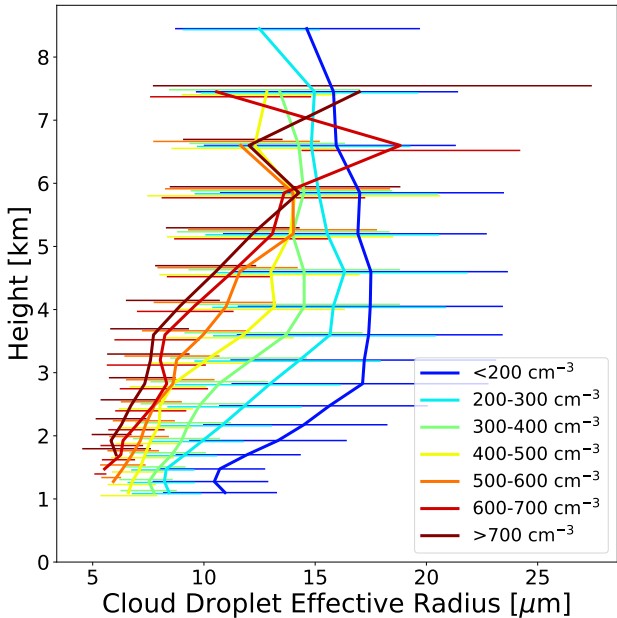

**Figure 3.** WRF-Chem simulated median cloud drop effective radius vertical profiles from all nested domain output during the study period, binned by below-cloud CCN concentration [cm$^{-3}$ at STP]. Error bars represent the 20th to 80th percentile for each level and are offset vertically for readability.





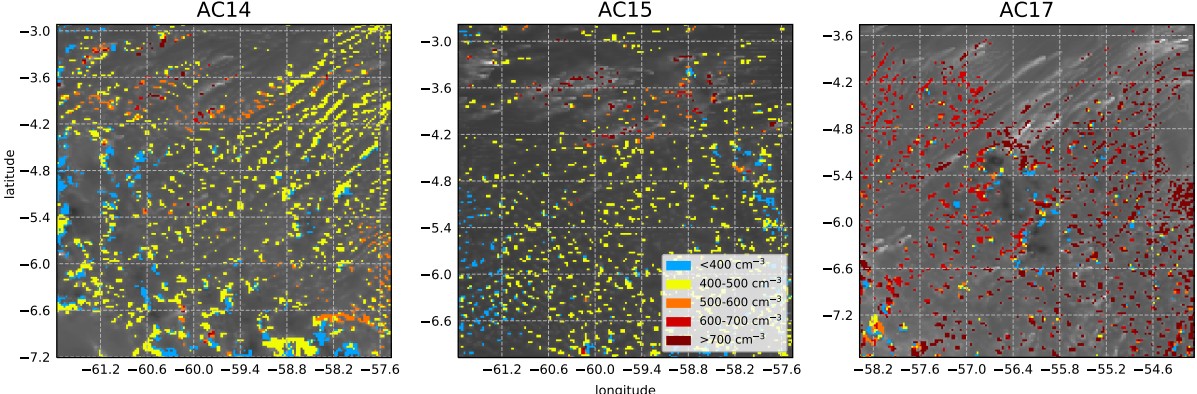

**Figure 4.** Spatial variability in modeled concentration of CCN at cloud base on three days (at 18Z) during the field campaign. Modelled aerosol optical depth (AOD) is shown as grey shading in the background, with brighter colors indicating higher AOD values.

quickly with height to a maximum $r_{\mathrm{eff}}$ under low CCN (clean) conditions, whereas under high CCN (polluted) conditions the radius does not reach a maximum until much higher in the atmosphere. The profiles reach a maximum and then remain roughly

constant at higher elevations. Under clean conditions, the maximum $r_{\mathrm{eff}}$ is larger and is reached at lower elevations. Profiles for the cleanest conditions also exhibit the largest maximum median $r_{\mathrm{eff}}$ of about 17 µm.

### 3.4 Comparison of modeled and observed $r_{\mathrm{eff}}$ profiles

WRF-Chem modelled $r_{\mathrm{eff}}$ profiles were compared to remote-sensed and in situ measured profiles. In Figure 4 we show snap-shots of the spatial variability of modeled CCN concentrations at cloud base for three different days. Figure 5 a-c then shows

$r_{\mathrm{eff}}$ profiles derived from specMACS from two-minute cloud scenes on these three days, below-cloud-CCN binned WRF $r_{\mathrm{eff}}$ profiles from three hours near the specMACS data collection time, and all in situ $r_{\mathrm{eff}}$ profile measurements within the nested model domain. Figure 5 d-f shows the known modeled and in situ CDNCs. Being a remote-sensing technique, no CDNC are available for the specMACS observations.

Note that this is an approximate comparison, as no exact colocation can be expected between in situ and remote-sensed

clouds, and we cannot compare individual modelled clouds directly to observed ones. Visual inspection of the slope and magnitude of median $r_{\mathrm{eff}}$ profiles measured by specMACS suggests that they match reasonably well to those from WRF-Chem, though in situ $r_{\mathrm{eff}}$ tend to be smaller than both the modeled or the ones retrieved by specMACS for all three cases investigated here.

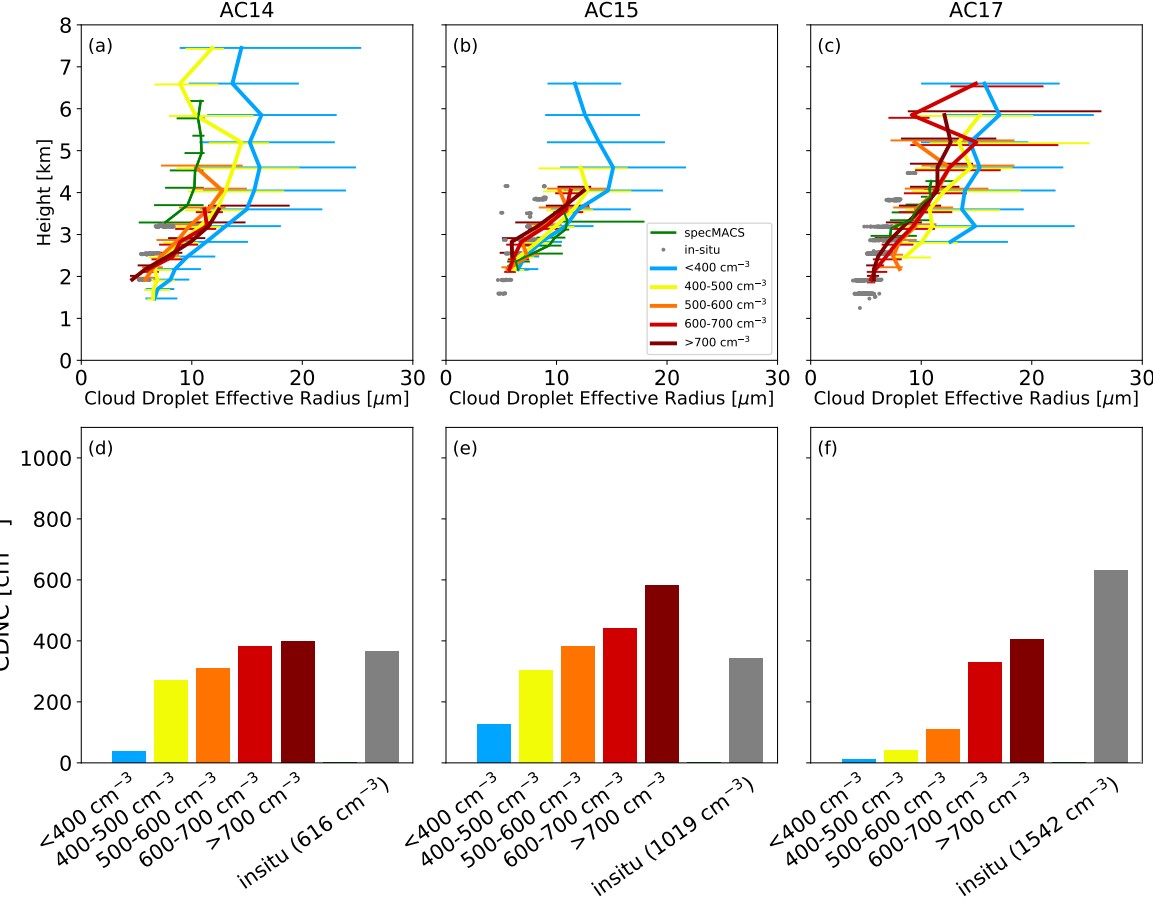

**Figure 5.** $r_{\mathrm{eff}}$ profiles and associated cloud droplet number concentration (CDNC) on three days during the field campaign. (a-c) show a comparison of median WRF-Chem, specMACS, and insitu $r_{\mathrm{eff}}$ profiles. (d-f) show the "true" below-cloud CCN-binned CDNC from WRF-Chem simulations and CDNC from in situ cloud profiling. Average in situ CCN concentrations (below 2 km) are presented in the bar label for the the in situ derived $N_{\mathrm{a}}$. See Section 3.1 for details regarding the definition of "average".

## 3.5 Number of activated cloud condensation nuclei at cloud base

As a more quantitative comparison of the different profiles, the number of activated CCNs at cloud base ($N_{\mathrm{a}}$) were derived for each profile based on the methodology proposed in Freud et al. (2011). For the same three days as in Figure 5, Figure 6 a-c shows the regressions between adiabatic liquid water content ($LWC_{\mathrm{a}}$) and mean volume radius ($r_{\mathrm{v}}$) that result (using Eq. 2) in the calculated $N_{\mathrm{a,calc}}$ values shown in Figure 6 d-f. $LWC_{\mathrm{a}}$ for the modeled profiles was calculated in model clouds at the same points as used for the $r_{\mathrm{eff}}$ values. For specMACS, a nested domain averaged $LWC_{\mathrm{a}}$ profile was used since the

below-cloud CCN is unknown for those measurements. The same profile was used for the in situ $LWC_{\mathrm{a}}$ to allow for direct comparisons. Only the increasing portion of the WRF-Chem profiles were used for the fits in Figure 6 a-c; points above the





first decrease that occurs above 4 km are excluded. The known CDNCs (Figure 5) and calculated $N_a$ (Figure 6) matched well given that CDNC is being viewed as equivalent to $N_a$, although $N_a$ is an upper limit for CDNC since CDNC can be influenced by processes like collision and coalescence. A direct comparison of the true and derived CDNC are shown in Figure 7. This

comparison demonstrates the effectiveness of the Freud et al. (2011) method for model data. The relationship is linear, but there is a systematic positive bias of derived CDNC. The factor of 0.7 as taken from the literature may be an underestimation for the modeled clouds. Sensitivity of the derivation to cloud base height may explain why using modeled $LWC_a$ resulted in high derived CDNC for two of the in situ derivations. Another contributor could be the high low-level CCN concentrations that were not reached in the model and in part by the use of an average model $LWC_a$ rather than a "true" $LWC_a$. Even though $N_{a,WRF}$

and $N_{a,calc}$ do not match exactly, general trends are captured. The $N_a$ derived from the specMACS $r_{eff}$ profiles ($N_{a,spec}$) fall within the range of modeled CDNCs (Figure 6 d-f). Compared to modeled CDNCs, specMACS-derived $N_{a,spec}$ are relatively high, low, and central for AC14, AC15, and AC17, respectively.

With the available data it is not possible to know the aerosol or below-cloud properties for the clouds sampled by specMACS. We suggest, however, that we can use the model results to deduce that the specMACS observed relatively polluted clouds during

AC14 (Figure 6 a,d), relatively clean clouds during AC15 (Figure 6 b,e), and medium polluted clouds during AC17 (Figure 6 c/f). The $N_a$ derived from the in situ profiles is higher than the others. While the calculated $N_a$ depends on the theoretical adabatic liquid water content ($LWC_a$), the measured LWC might in fact be lower. This finding should be explored further but is out of scope of this work.

## 3.6    Discussion

Modeled $r_{eff}$ tended to be larger than in situ measurements of $r_{eff}$. Subsequently, directly modeled and model-derived CNDC concentrations were lower than in situ measurements and derivations. Partly, these differences can be accounted for by the low modeled CCN concentrations (Figure 2). However, the $20^{th}$ to $80^{th}$ percentile range of modeled profiles with high below-cloud CCNs do overlap with the in situ data. The modeled $r_{eff}$ profiles began to saturate around $500 \, \text{cm}^{-3}$ at STP below-cloud CCN, with only small differences at higher concentrations (Figure 3), meaning that the modeled aerosol-cloud interactions saturate at

approximately that concentration. This concentration is well below the CCN concentrations characteristic of the dry season in the southern half of the Amazon Basin, which are typically in the range of 1000 to 7000 $\text{cm}^{-3}$ (Andreae et al., 2004; Andreae, 2009; Andreae et al., 2018). No such saturation was observed in the evaluation of modelled CDNC.

Increased model spatial resolution could potentially provide better agreement for these high-pollution situations, but a variety of hurdles (input data resolution of emissions and static data like land use, vegetation cover and topography, model formulation

of turbulence, statistical methods for output analysis) need to be overcome before reliable simulations at higher resolution are feasible. The horizontal grid resolution of 3 km is at the fine end of what regional modeling systems were designed for, reaching for 'terra incognita' (Wyngaard, 2004) in terms of resolution. Sensitivity simulations in which we simply increased the horizontal and/or vertical resolution by a factor of two did not lead to improved agreement with observations.

We deem our modeling study is representative for other regional scale chemistry-transport modeling studies of aerosol-cloud

interactions of convective clouds in situations strongly affected by biomass burning (e.g., Martins et al., 2009; Wu et al., 2011;



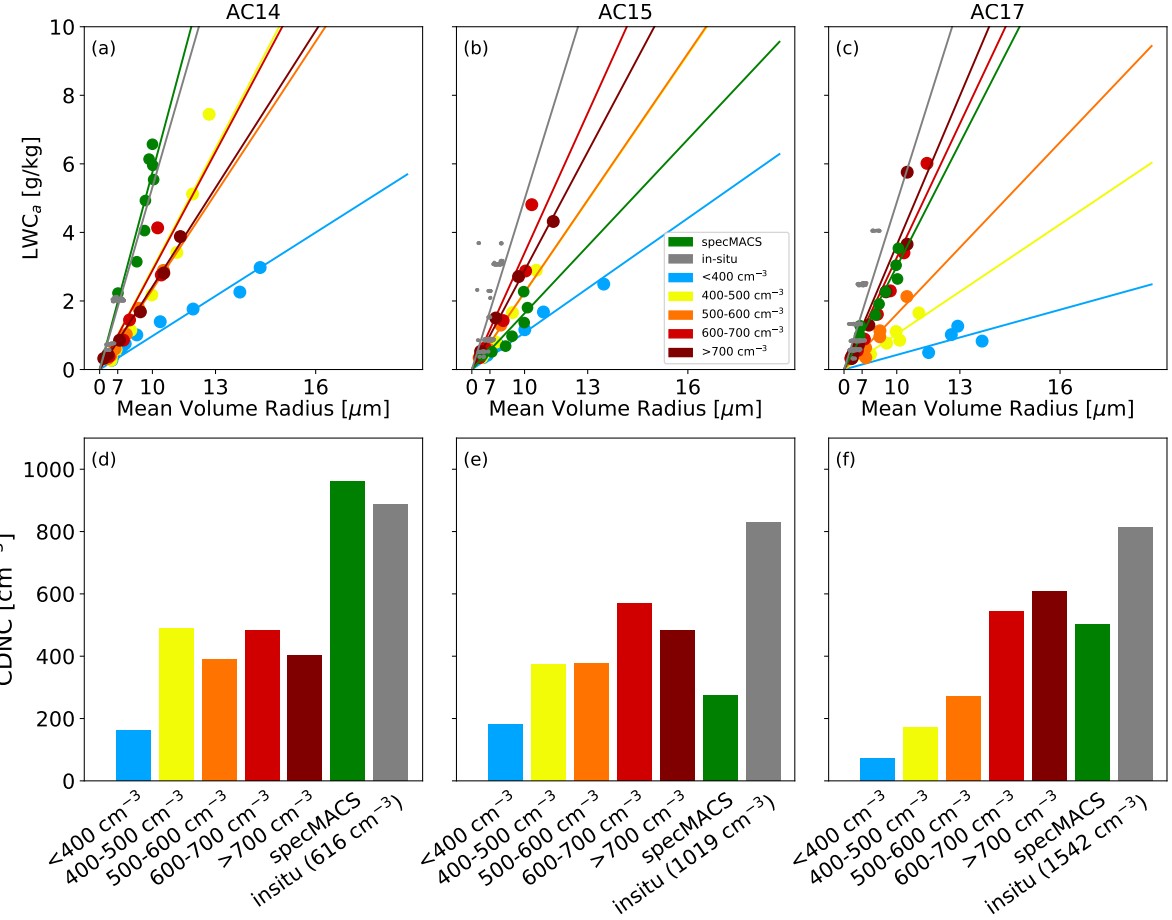

**Figure 6.** Derived cloud droplet number concentration (CDNC) on three days during the field campaign. (a-c) show the regressions between mean volume radius and adiabatic liquid water content ($LWC_a$) used to derive the CDNC as shown in (d-f). Average in situ CCN concentrations below 2 km are shown below the in situ derived $N_a$. (d-f) were derived from the slopes in (a-c), whereas Figure 5 d-f were more directly determined.

Archer-Nicholls et al., 2016). WRF-Chem is a widely used modeling system and similar to other regional modeling systems, our setup including a two-moment cloud microphysics scheme with a sectional aerosol module the current state of the art, as is the representation of aerosol-cloud interactions using the cloud activation scheme of Abdul-Razzak and Ghan (2000).

Comparisons between entire model domains and in situ measurements are inherently difficult since the exact measured 290 clouds will never be realistically measured due to the randomness of modeled clouds and the difference in scales. There are a variety of challenges involved with this comparison. However, especially at high CCN, the model overestimates $r_{\mathrm{eff}}$ and, therefore, underestimates $N_a$. The specMACS data experience similar comparison difficulties since each set only spans a cloud scene ($\sim$50 km) over a short time ($\sim$2 minutes). However, the retrieved $r_{\mathrm{eff}}$ profiles still fall within the in situ measurements





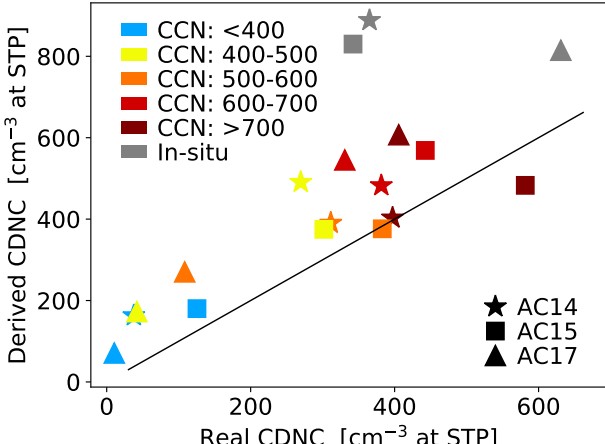

**Figure 7.** Comparison of real (i.e. CDP and CAS measured) CDNC with CDNC as derived using the Freud et al. (2011) method. Real CDNC for model data is average modeled CDNC in the model domain. Symbols indicate date, colors indicate model bin or in situ data. The one to one line is for reference. These are the same data as Figure 5 d-f and Figure 6 d-f.

and the model output. Profile values derived from specMACS measurements also tend to be smaller than the data from in situ
sampling, which is expected based on previous tests (Ewald et al., 2016).

We have demonstrated that the method by Freud et al. (2011) to derive cloud base CDNC from $r_{\mathrm{eff}}$ observations can successfully be applied in conjunction with simulated clouds to derive $N_{\mathrm{a}}$ from remotely sensed hyperspectral data of the specMACS instrument. The method is limited by its high sensitivity at low $N_{\mathrm{a}}$ due to the mathematical nature of the slope (i.e. steep slopes in Figure 6 a-c) and we are unable to verify its accuracy with the available data. It also uses an average mixing factor that may
vary for the cloud scenes measured my specMACS. However, using Figure 7 as a guide to the accuracy of the method, the uncertainties appear to be smaller than those from satellite retrievals, which are about 78 % at the pixel level (Grosvenor et al., 2018). We therefore propose that model results can be used to differentiate specMACS observations into clean and polluted conditions, which will need to be verified in future studies.

## 4   Conclusions

Aerosol-cloud interactions have been the focus of field campaigns and measurement development due to the large associated model uncertainty. Here we used novel observations taken on board of HALO during the ACRIDICON-CHUVA field campaign to evaluate cloud representation in a numerical model to aid in reducing this uncertainty. We demonstrated that we can reproduce realistic cloud properties (i.e., cloud droplet effective radius profiles) with a regional online-coupled chemistry-transport model at convection-permitting scales for the Amazon region during the biomass burning season. As expected by theory, the
number of CCN at cloud base has a major influence on cloud droplet size and the shape of the effective radius vertical profile. Increasing CCN leads to decreasing cloud droplet sizes, and we could show that both model and observations exhibit



quantitatively similar behavior. We also observed a saturation effect at high aerosol concentrations (number concentration of CCN larger than $500\,\mathrm{cm^{-3}}$ at STP) in the model, above which we find no further change in modelled effective droplet size or the shape of the droplet size profile. This model result is in disagreement with observations of microphysical effects at much

higher aerosol loading from previous campaigns (Reid et al., 1999; Andreae et al., 2004) and from the ACRIDICON-CHUVA campaign (Braga et al., 2017b). This finding casts doubt on the validity of regional scale modeling studies of the cloud albedo effect (Twomey, 1991) of convective clouds for biomass burning situations where the number concentration of CCN is larger than $500\,\mathrm{cm^{-3}}$ at STP.

*Code and data availability.* Model data, the source code used in the evaluation, as well as all observational data, are available from the
authors upon request.

*Author contributions.* PP made the simulations and conducted the analysis under the supervision of CK and TZ. PP, TZ and CK wrote the manuscript, with input from BM, MA, DR, RW and MW. MA, CP, MP, UP, DR, RW and MW contributed through fruitful discussions. FE, TKo, TJ, TKl, CM, SM, CP, MP, CV and RW provided measurements essential for this manuscript.

*Competing interests.* The authors declare no competing interests.

*Acknowledgements.* We thank the Leibniz Supercomputing Centre (LRZ) of the Bavarian Academy of Sciences and Humanities (BAdW) for the support and provisioning of computing infrastructure essential to this publication. We acknowledge use of the WRF-Chem preprocessor tools anthro_emiss, exo_coldens, fire_emiss, megan_bio_emiss, mozbc and wesely provided by the Atmospheric Chemistry Observations and Modeling Lab (ACOM) of NCAR. We acknowledge use of MOZART-4 global model output available at http://www.acom.ucar.edu/wrf-chem/mozart.shtml (last accessed 24.07.2018). We thank National Centers for Environmental Prediction for making Global Forecasting
System data publically available. We also thank DLR for access to the HALO database for all in situ data used in this study. This work was supported by the Max Planck Society, the DFG (Deutsche Forschungsgemeinschaft, German Research Foundation) Priority Program SPP 1294, the German Aerospace Center (DLR), the FAPESP (São Paulo Research Foundation) Grants 2009/15235-8 and 2013/05014-0, and a wide range of other institutional partners.



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
