# Peer review of "The challenge of simulating the sensitivity of the Amazonian clouds microstructure to cloud condensation nuclei number concentrations"

_Atmospheric Chemistry and Physics, 2019_

## Referee Comment (RC1) · Anonymous Referee #1 · 7 Aug 2019

This paper compares observed CCN, CDNC, and cloud droplet effective radius measured by the HALO aircraft in the Amazon with model simulations (WRF-Chem) and new remote sensing instrument deployed on HALO. This is an important topic, since measurements are needed to evaluate model predictions of cloud-aerosol interactions which are known to be highly uncertain, especially for convective clouds. The uncertainty in simulated cloud-aerosol-interactions impact predictions of aerosol indirect forcing in climate models as noted by past IPCC reports. In general, the results are presented logically and clearly although some addition description on the findings in some of the figures is needed. In addition, there are some flaws in the manuscript that need to be addressed before it is suitable for publication.

[Figure]

Major comments

1) The introductory material needs to be improved. The description of the relevant research on aerosol-cloud interactions is too brief. There needs to be more motivation here. For example in terms of a model, accurately simulating these interactions requires a good understanding and simulation of cloud and aerosol populations. So a quick summary of previous efforts to simulate these quantities in the Amazon would be useful as well. There have been some review articles on measuring and model cloud-aerosol-interactions that could have provided justification for the present work. In addition, the last three paragraphs seem to be more about methods than motivation for the research.

2) The authors note in several places comparisons with satellite derived droplet effective radius, but I could not find such comparisons. Either this needs to be included, or the words dropped from the manuscript. I would find it interesting to compare satellite derived values with in situ ones. I assume the satellite derived values assume vertically homogenous profiles, and it would be useful to compare HALO CCN profiles to test the validity of this assumption.

Specific Comments:

Line 8: The authors mention indirect effects here. But the paper never quantifies them as such. They do show CCN and droplet effective radii, but I would consider these simply consider these as parameters that are a metric (of many) of cloud-aerosol interactions. The indirect effect of biomass burning on clouds in a climate model sense is never discussed. So using these words in this way is misleading as to what the paper is about.

Line 11: The word "pollution" implies anthropogenic origin, but biomass burning is ambiguous in this case. Yes, the fires are probably started by humans, but is that the same as urban pollution? I see "highly polluted" used to describe high aerosol concentrations in the literature – in cases that are manmade or not.

Line 13: Here it states that simulated effective radii was too low, but later in Fig. 5 it looks to be higher than observed.

Line 20: Satellite retrievals are mentioned here, but as I noted elsewhere I did not see such as comparison. Do the authors mean specMACS which is remote sensing but on the HALO. There is some confusion here.

Line 36: The sentence should start as "Microphysical parameterizations . . ." to be more precise. The two papers cited in this sentence are not the best, since they are primarily about cloud microphysics and not cloud-aerosol interactions. I suggest including some of their more recent papers that focus more on this topic, as well as a few other authors.

Line 37: I do not think the Zhang et al. (2010) ever mentioned an improvement in terms of short-term forecasts. Instead, they demonstrated differences in the predictions associated with including such feedbacks. Either change this statement or find another paper that supports this claim.

Line 38: I would add precipitation to this list since it is an important meteorological forecast metric and its sensitivity to aerosol-cloud interactions has been examined by a number of studies.

Lines 41-43: I is not just high aerosol concentrations that provide the signals for aerosol-cloud interactions, it is more important to be in a situation with rapid changes in aerosol concentrations – from low to higher values. Aerosols can quickly "saturate" clouds so the high events listed here by themselves are not sufficient. One needs to see how a cloud responds when going between low and higher CCN.

Line 57: Table 1 probably does not need to be cited at this point. I assume that this should be cited somewhere in Section 2.2. It would also make more sense the table to be cited after Figure 1.

Lines 59-79: The description here seems to better fit the methods section. For the introductory material it would be better to state why a model is being used in conjunction

with the observations during the measurement campaign.

Lines 101: In terms of activation, is secondary activation included? This process may be important in deeper convection as described in Yang et al. (2015) and Fan et al. (2018). If not, ti would be useful to describe how it could influence the results in this study.

Line 113-114: What about clouds at the restart times? It takes some time for clouds to develop. Please comment on how that assumption affects the model simulations of aerosols.

Figure 1. Please include the grid spacing for both of the grids somewhere in the plot. Include a label for Manaus. Also label the outer nest in the figure itself and not just that caption. When looking at the figure initially, I assume the entire map was the outer domain.

Line 177: the title is good, however, the section does not provide a motivation as to why remote sensing and modeled cloud data are used when in situ data is available? I presume at this point, one would want to evaluate how well the remote sensing and modeled cloud data sets are, but that motivation is missing. After reading the rest of the manuscript, I cannot find any other use of satellite derived droplet effective radii. I was expecting a satellite vs in situ observation. Why is this being mention here then?

Line 180: The phrase "providing a valuable comparison" begs the question "in comparison with what?" I must be missing some point the authors are trying to make here. With the other two methods mentioned next?

Lines 207-209: I am not sure I agree with the assertion that the nested domains have a "homogeneous environment". Convective clouds can have complex organization, i.e. it is easily possible to have shallow clouds on one side of the domain and deep convection on the other, or clear skies in one region and cloudy in another, etc. Also the aerosols, largely from biomass burning are not necessarily uniform across the nested

domain. Can the authors provide some evidence regarding the homogeneous conditions over the nested domains?

Line 234: Some additional discussion of what is plotted in Figure 4 is needed. Presumably, only CCN at and below cloud base is shown. Presumably one can compute CCN everywhere and the authors just want to highlight it below cloud. But that is never really stated explicitly. Is the entire nested domain plotted? Again not clear. Also the AOD is very hard to see using the grey shading. Is there any other way to show the biomass burning plumes better? I can only really make them out for AC17. There is also no discussion of Figure 4 before jumping into Figure 5.

Lines 240-244: It is probably worth mentioning that the WRF-Chem droplet effective radii will depend on the specific microphysics scheme. One could argue that a spectral bin approach would be more realistic than a two moment approach, such as the Morrison scheme. Ideally the error bars on the modeled values is needed too – but that is impossible to quantify.

Lines 270-272: Have the authors evaluated the WRF-chem simulated size distributions with observations? Errors in the size distribution will affect estimates of CCN at various supersaturations. It is clear the simulated CCN is too low (Figure 2), but the simulated cloud droplet effective radii profiles are not that bad. There could be compensating errors in the model. Another comparison of observed vs simulated concentrations, using the AMS measurements on the HALO would provide some information about whether simulated aerosol concentrations are too low and whether the relative composition is correct (which will affect kappa). I am not saying an extensive evaluation is needed, but some additional discussion seems warranted on model performance. I appreciate the comments on resolution in the next paragraph, but as the authors stated I would expect a 3 km grid spacing to be adequate for this study.

Lines 314-318: This is a strong statement that is somewhat misleading. While I agree with the statement regarding microphysical effects at higher aerosol loading for the

studies listed, I believe there are other studies that do note a saturation effect (perhaps not for just biomass burning). The last sentence can only be applied to the particular model and its configuration for this study, rather than casting doubt on all regional scale modeling. The present model may be missing processes or has poor assumptions regarding other aerosol-cloud interactions, not to mention uncertainties in emissions, that affect the results. Other models may or may not have similar issues.

———————————————————

---

## Referee Comment (RC2) · Anonymous Referee #2 · 9 Aug 2019

This paper presents simulations with WRF-chem showing that it can reproduce trends in cloud droplet number concentration over the Amazon, although with a low bias. The model is also used to evaluate a parameterization of activated cloud condensation nuclei at cloud base, which is an important and interesting quantity. Some conclusions about the inability of regional modeling studies to represent aerosol-cloud interactions at high aerosol concentrations are drawn.

The paper uses interesting observations. Some are similar to those published already by Braga et al, but the specMACS observations are new and valuable. The model is state-of-the-art and has good potential to aid our understanding of the situation studied.

[Figure]

The evaluation of the Freud et al (2011) method is useful.

However, there are some significant shortcomings. Firstly, while the model evaluation in the paper is valuable, the authors need to do more to make the most of the excellent measurements available: measured and simulated in-situ aerosol concentrations should be compared, and it would also be useful to show simulated and observed liquid water content, even though in principle this is constrained by CDNC and effective radius. Secondly, and more importantly, the main conclusions of the paper are unconvincing, as I explain below. The paper will be suitable for publication in ACP if the authors are able to address my comments below.

**1   Major comments**

1. Can the authors explicitly compare simulated cloud-base aerosol or CCN concentrations to in-situ observations? Is it the aerosol concentration or the activation scheme/simulated updraft that explains why the model produces fewer CDNC than is observed? CPC, CCNC and UHSAS data are already published by Andreae et al (ACP 2018) so hopefully this is straightforward.

2. The introduction needs to put this study in the context of the relatively large body of literature relating specifically to aerosol-cloud interactions in the Amazon region and in deep convective clouds, which is currently hardly mentioned.

3. Maybe the authors thought this too obvious to be worth mentioning, but effective radius goes as $(q/N_d)^{1/3}$ where q is the liquid water content (see for example Morrison and Gettelman (2008)). Therefore a saturation-like behavior, or at least a strongly reduced dependence of $r_e$ on $N_d$, is expected for high $N_d$. For example, if $r_e$ is 10.0um when $N_d$ is 200cm$^{-3}$, $r_e$ is 6.9um at 600cm$^{-3}$, and 6.3um at 800cm$^{-3}$. So within uncertainties due to spatial fluctuations in liquid

water content, $r_e$ saturates at about 700cm$^{-3}$, while $N_d$ is still linearly increasing. Then, as in reality $N_d$ varies sub-linearly with activated CCN concentrations due to collision-coalesence, one would expect saturation in re as a function of CCN (or large Aitken and accumulation-mode aerosol concentrations) to happen even earlier. The authors should put the results in Section 3.4 in this context. Given that only very small changes in effective radius are expected as CCN increases, it is not clear that the saturation effect observed is unexpected. The results need to be put into this context.

4. Further to the previous comment, concerning the sentence 'The modeled r-eff profiles began to saturate around 500 cm-$3$ at STP below-cloud CCN, with only small differences at higher concentrations (Figure 3), meaning that the modeled aerosol-cloud interactions saturate at approximately that concentration.' While the effective radius is indeed the critical quantity that determines cloud albedo and the Twomey effect, it is cloud droplet number that determines the 'microphysical effects' of aerosols (on warm rain formation, droplet freezing rates, and droplet evaporation), and simulated CDNC apparently does not saturate (line 277). This apparent saturation of effective radius in the model is not sufficient grounds to say the model is in disagreement with observed aerosol-cloud microphysical interactions above 500/cc, as is stated in the conclusion. The statement that the validity of regional modeling studies of the Twomey effect (for which effective radius is the right variable) is in doubt also seems unfair at the moment. However, if the authors can show the saturation effect is still true when aerosol concentrations are doubled, or biomass burning emissions quadrupled, in a sensitivity study, then I think the statement could be better justified, at least for the authors' model.

5. Freud et al (2011) say effective radius is always larger than volumetric mean radius, not smaller, by an average of 8%, and one can also show $r_e > r_v$ for the gamma distributions used in the WRF microphysics schemes, so the equation at line 185 is the wrong way up.

**2 Minor comments**

In the introduction, Morrison and Gettelman (2008) is specified as the microphysics parameterization, while in the model description it is Morrison et al. (2009). I don't think these are the same, although I think they are both based on Morrison et al (2005). Please specify which is used.

L178: please add references to elucidate this statement. L184: Please split up this sentence, it currently seems to be two sentences joined together.

Figure 5: The CDNC is underestimated by the model while the effective radius is over-estimated, so the LWC might be simulated quite well, but it's hard to tell by eye. How does the LWC compare between model and observations?

The last part of this paper has some overlap with Braga et al, ACP 2017 (reference 'a' in the authors' notation), this is not a problem but it would be useful to discuss the overlap in the introduction and clarify that the study adds to Braga et al in that the Freud et al method is tested with a regional model.

A couple of strange sentences the authors may wish to fix: Abstract: "Our study casts doubt on the validity of regional scale modeling studies of the cloud albedo effect in convective situations for polluted situations….." (perhaps "convective, polluted situations"?) "Comparisons between entire model domains and in situ measurements are inherently difficult since the exact measured clouds will never be realistically measured….." ("measured….simulated?")

There are a few other typographical errors, "data" are plural, "less" is used in place of "fewer" but in general the written English is in good shape.

---

## Author Comment (AC1) · 6 Nov 2019

We thank both anonymous reviewers for their helpful comments. Below, we have answered all their remarks point-by-point, with the reviewers comments in black, our replies in blue, quotes from the manuscript in grey italic with changed text in red italic.

**Anonymous Referee #1 Received and published: 7 August 2019**

This paper compares observed CCN, CDNC, and cloud droplet effective radius measured by the HALO aircraft in the Amazon with model simulations (WRF-Chem) and new remote sensing instrument deployed on HALO. This is an important topic, since measurements are needed to evaluate model predictions of cloud-aerosol interactions which are known to be highly uncertain, especially for convective clouds. The uncertainty in simulated cloud-aerosol-interactions impact predictions of aerosol indirect forcing in climate models as noted by past IPCC reports. In general, the results are presented logically and clearly although some addition description on the findings in some of the figures is needed. In addition, there are some flaws in the manuscript that need to be addressed before it is suitable for publication.

**Major comments**

1) The introductory material needs to be improved. The description of the relevant research on aerosol-cloud interactions is too brief. There needs to be more motivation here. For example in terms of a model, accurately simulating these interactions requires a good understanding and simulation of cloud and aerosol populations. So a quick summary of previous efforts to simulate these quantities in the Amazon would be useful as well. There have been some review articles on measuring and model cloud-aerosol-interactions that could have provided justification for the present work. In addition, the last three paragraphs seem to be more about methods than motivation for the research.

We have significantly updated the introduction to address the reviewers comments. Due to the extent of the changes, instead of presenting the changes here, we would like to ask the reader to refer to the diffed manuscript attached to our responses.

2) The authors note in several places comparisons with satellite derived droplet effective radius, but I could not find such comparisons. Either this needs to be included, or the words dropped from the manuscript. I would find it interesting to compare satellite derived values with insitu ones. I assume the satellite derived values assume vertically homogenous profiles, and it would be useful to compare HALO CCN profiles to test the validity of this assumption.

Comparisons with satellites are mentioned when citing Rosenfeld et al. (2012), in which possibilities are explored to derive CCN from satellites, and when discussing the Freud et al. (2011) parameterisation, which could lend itself to be used to derive activated CCN at cloud base also from satellite observations.

We agree with the reviewer that this is a very interesting topic, but consider this out of scope for this work. Here we focus on data collected during the HALO campaign. To avoid further confusion we have removed language mentioning satellites from the abstract where it seemed unclear.

**Specific Comments:**

Line 8: The authors mention indirect effects here. But the paper never quantifies them as such. They do show CCN and droplet effective radii, but I would consider these simply consider these as parameters that are a metric (of many) of cloud-aerosol interactions. The indirect effect of biomass burning on clouds in a climate model sense is never discussed. So using these words in this way is misleading as to what the paper is about.

We have actually calculated the indirect effects (as top of atmosphere radiative forcing) and found it to be in line with previous studies. However, given our main findings regarding the inability of the model to represent very high CCN situations and their effects on cloud microphysics we refrained from adding those calculations to the manuscript. We have made the following modifications to the manuscript to accommodate the reviewers comments:

Indirect effects are mentioned three times in the manuscript. (1) In the abstract, where we clarify in the same sentence that we are discussing changes in the microphysics. We consider this to be appropriate. (2) In the introduction, where we motivate why our research is important. Also here we consider its use appropriate. (3) In the conclusions. We agree that the third mentioning could be considered unclear and have adapted it:

"[...] This finding casts doubt on the validity of using our setup for regional scale modeling studies of the cloud albedo effect (Twomey, 1991) of convective clouds for biomass burning situations at high CCN concentrations. Although we only tested one microphysics scheme, we demonstrated that a modern, complex parameterization does not imply accurate representation of cloud microphysical properties and suggest that calculations of the radiative forcing of these phenomena would therefore be unreliable. We conclude that there is a need for further model-measurement comparisons to better understand model biases."

**We have also added a paragraph to the discussion regarding the calculation of radiative forcing based on such a simulation:**

[...] in which we simply increased the horizontal and/or vertical resolution by a factor of two did not lead to improved agreement with observations.

Estimating the radiative forcing due to biomass burning is of central importance to evaluate its impact on the climate system. Calculating the top of atmosphere radiative forcing leads to an campaign average daytime cooling of -0.9 W m-2 (not shown), which is comparable to previous estimates and shows that our model behaves similar to existing studies. However, given the demonstrated lack of skill of the modeling system in representing the very strong CCN perturbations due to biomass burning, we refrained from further exploring their climate impacts.

We deem our modeling study is representative for other regional scale [...]"

Line 11: The word "pollution" implies anthropogenic origin, but biomass burning is ambiguous in this case. Yes, the fires are probably started by humans, but is that the same as urban pollution? I see "highly polluted" used to describe high aerosol concentrations in the literature – in cases that are manmade or not.

We do see how this might be implied by the reader when reading the word 'pollution', but think that it becomes clear that it is the contamination of the air with trace gases and particles by wildfires that is of concern in this manuscript. The definition of pollution (Oxford Advanced Learners Dictionary, Cambridge Dictionary) does not mention "pollution" necessarily being of anthropogenic origin. Given the lack of suitable alternatives and suggestions by the reviewer we would prefer to stick with our original solution.

Line 13: Here it states that simulated effective radii was too low, but later in Fig. 5 it looks to be higher than observed.

**This should indeed say overestimation and has been corrected.**

Line 20: Satellite retrievals are mentioned here, but as I noted elsewhere I did not see such as comparison. Do the authors mean specMACS which is remote sensing but on the HALO. There is some confusion here.

**See our answer above.**

Line 36: The sentence should start as "Microphysical parameterizations ..." to be more precise. The two papers cited in this sentence are not the best, since they are primarily about cloud microphysics and not cloud-aerosol interactions. I suggest including some of their more recent papers that focus more on this topic, as well as a few other authors.

The sentence has been revised and includes more authors. It now reads:

"[...] Cloud microphysical parameterizations with varying levels of complexity have been incorporated into numerical models of the atmosphere (e.g., Khain and Sednev, 1996; Morrison et al., 2005; Seifert and Beheng, 2006; Grützun et al., 2008; Thompson and Eidhammer, 2014), which provides opportunities to better understand the underlying physical processes. [...]"

Line 37: I do not think the Zhang et al. (2010) ever mentioned an improvement in terms of short-term forecasts. Instead, they demonstrated differences in the predictions associated with including such feedbacks. Either change this statement or find another paper that supports this claim.

**The statement has been changed and the citation is no longer included.**

Line 38: I would add precipitation to this list since it is an important meteorological forecast metric and its sensitivity to aerosol-cloud interactions has been examined by a number of studies.

**Done.**

Lines 41-43: I is not just high aerosol concentrations that provide the signals for aerosol-cloud interactions, it is more important to be in a situation with rapid changes in aerosol concentrations – from low to higher values. Aerosols can quickly "saturate" clouds so the high events listed here by themselves are not sufficient. One needs to see how a cloud responds when going between low and higher CCN.

Though this comment raises an interesting question, the authors believe the rate of change of CCN cannot be investigated in the context of this study due to limitations in the temporal resolution of the in-situ and remotely sensed data. Additionally, in this study individual clouds do not typically pass between different CCN regimes - in part due to relatively low spatial gradients of CCNs and in part due to limitations in spatial resolution of model. Instead, different clouds are are influenced by different CCN concentrations across the region as demonstrated in Figure 4.

Line 57: Table 1 probably does not need to be cited at this point. I assume that this should be cited somewhere in Section 2.2. It would also make more sense the table to be cited after Figure 1.

Table 1 and Figure 1 are still cited together, but now, as suggested, in the Methods section, where the field campaign is described.

Lines 59-79: The description here seems to better fit the methods section. For the introductory material it would be better to state why a model is being used in conjunction with the observations during the measurement campaign.

Much of the description from the late part of the introduction has been incorporated into the methods section.

Lines 101: In terms of activation, is secondary activation included? This process may be important in deeper convection as described in Yang et al. (2015) and Fan et al. (2018). If not, it would be useful to describe how it could influence the results in this study.

As the innermost domain - the domain all results shown in the manuscript are based on - is considered to be convection permitting, no convection parameterisation is active. Hence, 'secondary activation' understood as the activation of cloud condensation nuclei that were entrained through turbulent mixing at cloud sides is considered to the extent that entrainment is accurately simulated. Similar, 'secondary activation' within clouds due to local supersaturation is considered to the extent that the model can represent the additional local in-cloud supersaturation at the grid scale.

We agree that this topic should be noted in the text, and have added language to the methods section:

[...] of the cloud droplets in which they are incorporated in, including processes like washout from precipitation or re-evaporation. Secondary, in-cloud activation of aerosol particles to cloud droplets is only considered to the extent that entrainment and in-cloud supersaturation is represented on the grid-scale. Cloud chemistry and limited heterogeneous processes are included as [...]"

Line 113-114: What about clouds at the restart times? It takes some time for clouds to develop. Please comment on how that assumption affects the model simulations of aerosols.

As described in our methods section, the outer domain is run continuously and does not have restarts / gaps. We start the nested domain a couple of hours before the plane takes off, which represents early morning (local time). Hence, we allow the diurnal cycle of convection to be represented at the fine scale from this time onwards. We comment that small convective systems that might live through the night will not be capture by the model at the fine scale, rather will they be treated by the coarser, outer domain and hence merely approximated by the convection parameterization.

Figure 1. Please include the grid spacing for both of the grids somewhere in the plot. Include a label for Manaus. Also label the outer nest in the figure itself and not just that caption. When looking at the figure initially, I assume the entire map was the outer domain.

We agree that these suggestions allow for easier interpretation of the figure. We have added labels to better identify the outer model domain and the city of Manaus. Plotting the grid spacing would be illegible, but this sentence was added to the figure caption: The outer domain resolution is 15 km and the inner domain resolution is 3 km.

Line 177: the title is good, however, the section does not provide a motivation as to why remote sensing and modeled cloud data are used when in situ data is available? I presume at this point, one would want to evaluate how well the remote sensing and modeled cloud data sets are, but that motivation is missing. After reading the rest of the manuscript, I cannot find any other use of

satellite derived droplet effective radii. I was expecting a satellite vs in situ observation. Why is this being mention here then?

More motivation has been added to the introduction. Satellite data are not used in this study and text has been changed to reduce confusion. The reviewer is asked to refer to our reply above for further clarification.

Line 180: The phrase "providing a valuable comparison" begs the question "in comparison with what?" I must be missing some point the authors are trying to make here. With the other two methods mentioned next?

"[...] providing a valuable comparison [...]"has been changed to"[...] providing a valuable comparison to remotely sensed and modeled data [...]"

Lines 207-209: I am not sure I agree with the assertion that the nested domains have a "homogeneous environment". Convective clouds can have complex organization, i.e. it is easily possible to have shallow clouds on one side of the domain and deep convection on the other, or clear skies in one region and cloudy in another, etc. Also the aerosols, largely from biomass burning are not necessarily uniform across the nested domain. Can the authors provide some evidence regarding the homogeneous conditions over the nested domains?

Figure 4 gives an indication of the spatial variability in CCN at cloud base, as well as the typical size of the clouds within the model domain. We agree with the reviewer that there is some heterogeneity within the small domain, but consider this to be inevitable with this kind of modeling approach. As convection is stochastic we need to apply a statistical comparison anyway, and we try to capture potential variability in cloud conditions by using large samples from the model result and adding appropriate error calculations.

Line 234: Some additional discussion of what is plotted in Figure 4 is needed. Presumably, only CCN at and below cloud base is shown. Presumably one can compute CCN everywhere and the authors just want to highlight it below cloud. But that is never really stated explicitly. Is the entire nested domain plotted? Again not clear. Also the AOD is very hard to see using the grey shading. Is there any other way to show the biomass burning plumes better? I can only really make them out for AC17. There is also no discussion of Figure 4 before jumping into Figure 5.

In the figure caption, we have clarified that the plotted region is the entire nested domain, and that CCN concentrations are only shown where clouds existed.

"Spatial variability in modeled concentration of CCN at cloud base on three days (at 18Z) for the entire nested domain. Modelled aerosol optical depth (AOD) is shown as grey shading in the background, with brighter colors indicating higher AOD values. CCN concentrations are only shown where clouds were present." We also added this text to the discussion below:

"[...]This figure demonstrates the influence of the fires on the regional CCN concentrations and highlights the CCN variability at large and small scales. Three dimensional CCN fields were simulated, but below-cloud concentrations (i.e. CCN concentration below the lowest cloudy point in a column) are most relevant for cloud droplet size.[...]"

**We have finally also increased axis label sizes for readability.**

Lines 240-244: It is probably worth mentioning that the WRF-Chem droplet effective radii will depend on the specific microphysics scheme. One could argue that a spectral bin approach would be more realistic than a two moment approach, such as the Morrison scheme. Ideally the error bars on the modeled values is needed too – but that is impossible to quantify.

We agree that spectral bin microphysics might be of interest for future investigation. The following paragraph has been added to the discussion section:

**"[...]**

More complex parameterizations of cloud microphysics, such as spectral bin microphysics (e.g. Grützun et al., 2008; Khain and Sednev, 1996), have been developed and used before in case studies. Such more complex parameterisations might improve the representation of the cloud droplet size spectra and hence also modelled reff. Such parameterisations are, however, still computationally too expensive to be used on a regular basis or in the context of a climate study. [...]"

Lines 270-272: Have the authors evaluated the WRF-chem simulated size distributions with observations? Errors in the size distribution will affect estimates of CCN at various supersaturations. It is clear the simulated CCN is too low (Figure 2), but the simulated cloud droplet effective radii profiles are not that bad. There could be compensating errors in the model. Another comparison of observed vs simulated concentrations, using the AMS measurements on the HALO would provide some information about whether simulated aerosol concentrations are too low and whether the relative composition is correct (which will affect kappa). I am not saying an extensive evaluation is needed, but some additional discussion seems warranted on model performance. I appreciate the comments on resolution in the next paragraph, but as the authors stated I would expect a 3 km grid spacing to be adequate for this study.

We have evaluated our simulations against a range of observations taken by the HALO research aircraft during ACRIDICON-CHUVA, including CCN and AMS observations. Regarding AMS observations, we have found satisfactory performance, with good agreement with observations of the relative contribution of components that affect kappa (especially sulfate and

organics) as well as the total PM 1 non-refractory mass (not shown). Shown below is the evaluation plot for CCN using CAS-DPOL data:

Figure: Normalized PDFs of insitu CAS-DPOL and modeled WRF-Chem track cloud drop number concentration (CDNC) data from the entire inner domain. A direct comparison of measurements and model output is not feasible, because the modeled clouds do not occur at the same place and time as those in reality. The modeled and measured CDNC agree reasonably well, but do not reach extreme values above about 1000 CDNC.

Clearly, there is model skill for most situations, but also a deficiency visible in representing very high CCN concentrations.

Lines 314-318: This is a strong statement that is somewhat misleading. While I agree with the statement regarding microphysical effects at higher aerosol loading for the studies listed, I believe there are other studies that do note a saturation effect (perhaps not for just biomass burning). The last sentence can only be applied to the particular model and its configuration for this study, rather than casting doubt on all regional scale modeling. The present model may be missing processes or has poor assumptions regarding other aerosol-cloud interactions, not to mention uncertainties in emissions, that affect the results. Other models may or may not have similar issues.

**We realise this statement has been too broad, and have adapted it to read:**

"[...] above which we find no further change in modelled effective droplet size or the shape of the droplet size profile. **Our** model results are in disagreement with observations of microphysical

effects at much higher aerosol loading from previous campaigns (Reid et al., 1999; Andreae et al., 2004) and from the ACRIDICON-CHUVA campaign (Braga et al., 2017b). This finding casts doubt on the validity of using a setup like ours for regional scale modeling studies of the cloud albedo effect (Twomey, 1991) of convective clouds for biomass burning situations at high CCN concentrations. Although we only tested one microphysics scheme, we demonstrated that a modern, complex parameterization does not imply accurate representation of cloud microphysical properties and suggest that calculations of the radiative forcing of these phenomena would therefore be unreliable. We conclude that there is a need for further model-measurement comparisons to better understand model biases."

**Anonymous Referee #2 Received and published: 9 August 2019**

This paper presents simulations with WRF-chem showing that it can reproduce trends in cloud droplet number concentration over the Amazon, although with a low bias. The model is also used to evaluate a parameterization of activated cloud condensation nuclei at cloud base, which is an important and interesting quantity. Some conclusions about the inability of regional modeling studies to represent aerosol-cloud interactions at high aerosol concentrations are drawn. The paper uses interesting observations. Some are similar to those published already by Braga et al, but the specMACS observations are new and valuable. The model is state-of-the-art and has good potential to aid our understanding of the situation studied.

The evaluation of the Freud et al (2011) method is useful. However, there are some significant shortcomings. Firstly, while the model evaluation in the paper is valuable, the authors need to do more to make the most of the excellent measurements available: measured and simulated in-situ aerosol concentrations should be compared, and it would also be useful to show simulated and observed liquid water content, even though in principle this is constrained by CDNC and effective radius. Secondly, and more importantly, the main conclusions of the paper are unconvincing, as I explain below. The paper will be suitable for publication in ACP if the authors are able to address my comments below.

**Major comments**

1. Can the authors explicitly compare simulated cloud-base aerosol or CCN concentrations to in-situ observations? Is it the aerosol concentration or the activation scheme/simulated updraft that explains why the model produces fewer CDNC than is observed? CPC, CCNC and UHSAS data are already published by Andreae et al (ACP 2018) so hopefully this is straightforward.

We have evaluated model performance against observations of aerosol properties / CCN concentrations and found good agreement (see also our replies to reviewer 1), but also see some deficiencies for high aerosol concentrations. An evaluation plot has been added as part of a new supplement to the paper, showing comparisons against CAS-DPOL measurements. We agree with the reviewer that it can be both, the aerosol concentrations as well as the activation scheme that is responsible for the deficiencies found. In our current setup we are not able to disentangle those effects, and we think this is adequately represented in the manuscript.

2. The introduction needs to put this study in the context of the relatively large body of literature relating specifically to aerosol-cloud interactions in the Amazon region and in deep convective clouds, which is currently hardly mentioned.

The second half of the introduction has been changed substantially, including references to previous work in the Amazon. The reviewer is referred to the diffed version of the manuscript to evaluate the changes made.

3. Maybe the authors thought this too obvious to be worth mentioning, but effective radius goes as (q/Nd)1/3 where q is the liquid water content (see for example Morrison and Gettelman (2008)). Therefore a saturation-like behavior, or at least a strongly reduced dependence of re on Nd, is expected for high Nd. For example, if re is 10.0um when Nd is 200cm–3, re is 6.9um at 600cm–3, and 6.3um at 800cm–3. So within uncertainties due to spatial fluctuations in liquid water content, re saturates at about 700cm–3, while Nd is still linearly increasing. Then, as in reality Nd varies sub-linearly with activated CCN concentrations due to collision-coalesence, one would expect saturation in re as a function of CCN (or large Aitken and accumulation-mode aerosol concentrations) to happen even earlier. The authors should put the results in Section 3.4 in this context. Given that only very small changes in effective radius are expected as CCN increases, it is not clear that the saturation effect observed is unexpected. The results need to be put into this context.

The authors are aware that the effective radius theoretically saturates, but agree that it should be explicitly stated. We have added this text to section 3.4:

"[...] The relatively small differences between reff profiles at larger CDNC are expected because the theoretical relationship between  $r_{eff}$  and CDNC is  $r_{eff} \sim (LWC/CDNC)^{(\%)}$  (Morrison and Gettleman, 2008). A linear relationship between LWC and CDNC therefore results in saturation of  $r_{eff}$ . However, at what CDNC this saturation occurs is not equally well described. [...]"

4. Further to the previous comment, concerning the sentence 'The modeled r-eff profiles began to saturate around 500 cm-3 at STP below-cloud CCN, with only small differences at higher concentrations (Figure 3), meaning that the modeled aerosol-cloud interactions saturate at approximately that concentration.' While the effective radius is indeed the critical quantity that determines cloud albedo and the Twomey effect, it is cloud droplet number that determines the 'microphysical effects' of aerosols (on warm rain formation, droplet freezing rates, and droplet evaporation), and simulated CDNC apparently does not saturate (line 277). This apparent saturation of effective radius in the model is not sufficient grounds to say the model is in disagreement with observed aerosol-cloud microphysical interactions above 500/cc, as is stated in the conclusion.

The conclusion and discussion have been adapted so that the distinction between the saturation of cloud albedo / Twomey effect vs. microphysical effects are clear. As there have been

numerous adaptations to the text, the reviewer is referred to the manuscript to review the changes made.

The statement that the validity of regional modeling studies of the Twomey effect (for which effective radius is the right variable) is in doubt also seems unfair at the moment. However, if the authors can show the saturation effect is still true when aerosol concentrations are doubled, or biomass burning emissions quadrupled, in a sensitivity study, then I think the statement could be better justified, at least for the authors' model.

We can indeed show that there is no further appreciable change in reff when double biomass burning emissions, as shown in the figure below, where we re-ran case AC17 (2014-09-27) with twice the emissions from biomass burning:

---

## Referee Report (RR1)

This paper documents a modeling study of CCN in the Amazon with WRF-Chem, evaluated with ACRIDICON-CHUVA observations. The authors find biomass burning aerosols influence the Amazon clouds, but also suggest a saturation of the effect in very polluted conditions.

The authors have gone some way towards addressing the most important of my previous comments. The review responses were reasonably comprehensive and well organized. Both the introduction and conclusion of the paper are improved. However, the paper text still needs some important changes before it is suitable for publication.

**Major comments**
Abstract: The authors change "underestimation" of CDNC to overestimation, but Figure 2 hasn't changed, and still shows that the model underestimates CDNC. I am confused as to why this was changed and why the abstract says the slope is two when it is 0.334.

The last sentence of the abstract still needs changing, in line with the modified conclusions (but see below).

I did not find the promised supplement.

In my previous review, I said:
"While the effective radius is indeed the critical quantity that determines cloud albedo and the Twomey effect, it is cloud droplet number that determines the 'microphysical effects' of aerosols (on warm rain formation, droplet freezing rates, and droplet evaporation), and simulated CDNC apparently does not saturate (line 277). This apparent saturation of effective radius in the model is not sufficient grounds to say the model is in disagreement with observed aerosol-cloud microphysical interactions above 500/cc, as is stated in the conclusion."

I don't feel this comment has been adequately addressed. The authors claim to have separated Twomey effects from microphysical effects, but they only do this in the discussion, not in the abstract or the conclusions. The authors still say "the additional CCN emitted from local fires did not cause a notable change in modelled cloud microphysical properties" in their abstract, but the additional CCN clearly leads to increased CDNC – which is an obvious and observed change in microphysical properties. Again in the conclusions, the authors say "Our model results are in disagreement with observations of microphysical effects at much higher aerosol loading from previous campaigns", and this statement is not at all justified. The simulations clearly do show microphysical effects, but they may not be the same effects as the microphysical effects observed.

In fact, in polluted conditions, if CDNC increases when aerosol concentrations increase, while r-eff does not increase, that means LWC must increase, because of the relation r-eff $\sim(LWC/N)^{0.333}$. Increasing LWC with increasing CDNC is an aerosol-cloud microphysical interaction, in fact one quite commonly observed in models (e.g. McCoy et al, ACP 2018), and not a saturation of anything. Because it probably is not the same aerosol-cloud interaction as seen in observations, the structure of the paper may not need changing. **However, I really do think the authors should make a much more fundamental change to their conclusions than they did in response to my previous review**. The saturation of the Twomey effect in polluted conditions due to the saturation of effective radius is obvious because of the re$\sim$ $(LWC/N)^{0.33}$ relation, and adds nothing new to our understanding of atmospheric science. On the other hand, other findings in the paper, for example, the testing of the Freud parameterization, the finding that CDNC is underestimated, are legitimate new findings that are worth publishing. The conclusion should be rewritten to emphasise these instead, and the abstract changed to match.

**Minor comments**

The Reid et al, 1999 paper is highly relevant to this study and should not be brought up for the first time in the conclusion. The authors should discuss the main findings of the paper in the introduction, and put their results more fully in the context of Reid's work in the discussion.

L37 "nucleii"->"nuclei"
L371 "measured my"->"measured by"

---

## Author Response (AR2)

We thank both anonymous reviewers for their helpful comments. Below, we have answered all their remarks point-by-point, with the reviewers comments in black, our replies in blue, *quotes from the manuscript in grey italic* **with changed text in red italic**.

**Anonymous Referee #1**

The authors have addressed my comments sufficiently so that I find it suitable for publication, with one exception.

The introduction has been greatly improved; however, there is a reference that should be added that is relevant to this study. In the sentence "However, few studies have attempted to combine analysis of regional numerical models with measurements (Ten Hoeve et al., 2011)" in line 59 should also include a reference to Fan et al. Science (2018). This paper also uses a regional model combined with measurements to look at cloud-aerosol interactions.

A reference to Fan et al. (2018) has been added.

That paper also argues that secondary activation can occur in deep convection from aerosols entrained at cloud base. Some of those aerosols (small) may not be activated until higher altitudes where the supersaturation is different than at cloud base. Polonik included some text in the revision on secondary activation (which I encouraged them to do) that implied that it occurs via lateral mixing, but that is not the only means of secondary activation.

We have added more text to highlight the potential further effects on cloud microphysics do to the activation of ultrafine particles.

*"[...] Secondary, in-cloud activation of aerosol particles to cloud droplets is only considered to the extent that entrainment and in-cloud supersaturation is represented on the grid-scale.* **Other sources of secondary activation such as ultrafine particles (Fan et al., 2018) are not considered.** *Cloud chemistry and limited heterogeneous processes are included as [...]"*

**Anonymous Referee #2**

This paper documents a modeling study of CCN in the Amazon with WRF-Chem, evaluated with ACRIDICON-CHUVA observations. The authors find biomass burning aerosols influence the Amazon clouds, but also suggest a saturation of the effect in very polluted conditions.
The authors have gone some way towards addressing the most important of my previous comments. The review responses were reasonably comprehensive and well organized. Both the introduction and conclusion of the paper are improved. However, the paper text still needs some important changes before it is suitable for publication.

**Major comments**

Abstract: The authors change "underestimation" of CDNC to overestimation, but Figure 2 hasn't changed, and still shows that the model underestimates CDNC. I am confused as to why this was changed and why the abstract says the slope is two when it is 0.334.

We thank the reviewer for carefully re-reading the manuscript. This was an erroneous edit. The model does indeed underestimate CDNC, just as the reviewer remarked. We have corrected this and ask the reviewer to refer to the updated abstract in our reply to your next comment.

The last sentence of the abstract still needs changing, in line with the modified conclusions (but see below).

The reviewer is referred to our answers to his/her comments below. The updated abstract now reads:

*"[...] on cloud microphysical **and optical** properties (droplet number concentration and effective radius).*
*We found agreement between modeled and observed median cloud droplet number concentrations (CDNC) for low values of CDNC, i.e., low levels of pollution. In general, a linear relationship between modelled and observed CDNC with a slope of **0.3** was found, which **implies a systematic underestimation** of modeled CDNC as compared to measurements. Variability in cloud condensation nuclei (CCN) number concentrations **was also underestimated** and cloud droplet effective radii ($r_{eff}$) **were overestimated** by the model. Modeled effective radius profiles began to saturate around 500 CCN per $cm^3$ at cloud base, indicating an upper limit for the model sensitivity well below CCN concentrations reached during the burning season in the Amazon Basin. **Additional** CCN emitted from local fires did not cause a notable change in modelled cloud **droplet effective radii. Finally, we also** evaluate a parameterization of CDNC at cloud base using more readily available cloud microphysical properties, **showing that we are able to derive CDNC at cloud base from cloud-side remote sensing observations.**"*

I did not find the promised supplement.

The reviewer is correct, we mistakenly referenced a supplement that we ultimately deemed unnecessary. The evaluation plot mentioned was actually directly included in the responses to reviewers, and is reproduced here below:

[Figure]

Figure: Normalized PDFs of insitu CAS-DPOL and modeled WRF-Chem track cloud drop number concentration (CDNC) data from the entire inner domain. A direct comparison of measurements and model output is not feasible, because the modeled clouds do not occur at the same place and time as those in reality. The modeled and measured CDNC agree reasonably well, but do not reach extreme values above about 1000 per cm$^3$.

In my previous review, I said:

"While the effective radius is indeed the critical quantity that determines cloud albedo and the Twomey effect, it is cloud droplet number that determines the 'microphysical effects' of aerosols (on warm rain formation, droplet freezing rates, and droplet evaporation), and simulated CDNC apparently does not saturate (line 277). This apparent saturation of effective radius in the model is not sufficient grounds to say the model is in disagreement with observed aerosol-cloud microphysical interactions above 500/cc, as is stated in the conclusion."

I don't feel this comment has been adequately addressed. The authors claim to have separated Twomey effects from microphysical effects, but they only do this in the discussion, not in the abstract or the conclusions. The authors still say "the additional CCN emitted from local fires did not cause a notable change in modelled cloud microphysical properties" in their abstract, but the additional CCN clearly leads to increased CDNC – which is an obvious and observed change in microphysical properties.

The reviewer is correct, the addition of biomass burning resulted in negligible changes in cloud droplet effective radii, not cloud microphysical properties as originally stated. This has been corrected, please refer to our reply above for a reproduction of the updated abstract.

Again in the conclusions, the authors say "Our model results are in disagreement with observations of microphysical effects at much higher aerosol loading from previous campaigns", and this statement is not at all justified. The simulations clearly do show microphysical effects, but they may not be the same effects as the microphysical effects observed.

We agree with the reviewer that this statement is too general. It has been amended, and the reviewer is referred to our updated conclusion reproduced below these comments for reference.

In fact, in polluted conditions, if CDNC increases when aerosol concentrations increase, while r-eff does not increase, that means LWC must increase, because of the relation r-eff ~(LWC/N)^0.333. Increasing LWC with increasing CDNC is an aerosol-cloud microphysical interaction, in fact one quite commonly observed in models (e.g. McCoy et al, ACP 2018), and not a saturation of anything.
Because it probably is not the same aerosol-cloud interaction as seen in observations, the structure of the paper may not need changing. However, I really do think the authors should make a much more fundamental change to their conclusions than they did in response to my previous review. The saturation of the Twomey effect in polluted conditions due to the saturation of effective radius is obvious because of the re~(LWC/N)^0.33 relation, and adds nothing new to our understanding of atmospheric science.

We agree that corresponding changes of CDNC, LWC, and r-eff are microphysical interactions. Stil, we find that the modelled r-eff profile does not change anymore above 500 per cm$^3$, which is frequently even below the observed regional background, and this has clear implications (no further Twomey effect due to biomass burning emissions) for the radiative impact of clouds. We agree that this saturation is not a new finding, but consider it important to show that this occurs in the model (already) at CCN roughly above 500 per cm$^3$.  It is actually vital to be able to determine the point (in terms of LWC and CDNC) at which this happens and whether model and observations agree, as this determines the ability of the model to accurately represent the radiative impact of biomass burning events through the cloud-albedo feedback.We have therefore adapted the text accordingly (see below).

On the other hand, other findings in the paper, for example, the testing of the Freud parameterization, the finding that CDNC is underestimated, are legitimate new findings that are worth publishing. The conclusion should be rewritten to emphasise these instead, and the abstract changed to match.

We have changed the wording on this topic to state our findings without making them sound surprising. We have also softened the language about the implications for other modeling studies. However, we clearly demonstrate that there is a systematic overestimation of effective radii in the model compared to in situ measurements, which would translate directly into an underestimation of cloud reflectivity. As suggested, more emphasis has been placed on the new application of the parameterization. The updated conclusions now read:

*"Aerosol-cloud interactions have been the focus of field campaigns and measurement development due to the large associated model uncertainty. Here we used novel observations taken on board the HALO aircraft during the ACRIDICON-CHUVA field campaign to evaluate cloud representation in a numerical model to* **help reduce** *this uncertainty. We demonstrated that we can reproduce realistic cloud properties (i.e., cloud droplet effective radius profiles) with a regional online-coupled chemistry-transport model at convection-permitting scales for the Amazon region during the biomass burning season.*

*As expected* **from** *theory, the number of CCN at cloud base has a major influence on cloud droplet size and the shape of the* **vertical profile of cloud droplet effective radius**. *Increasing CCN leads to decreasing cloud droplet sizes, and we* **demonstrated that the model and the** *observations exhibit quantitatively similar behavior. We also observed a saturation effect at high aerosol concentrations* **in the model** *(number concentration of CCN larger than 500 $cm^{-3}$ at STP), above which we find no further change in modelled effective droplet size or the shape of the droplet size profile.* **Observations** *from previous campaigns (Reid et al., 1999; Andreae et al., 2004) and from the ACRIDICON-CHUVA campaign (Braga et al., 2017b)* **have demonstrated substantial Twomey effects at much higher aerosol loadings. Additionally, the relation between modelled and observed CDNC is linear and has a slope of 0.3, indicating a considerable underestimation of cloud droplet number concentrations by the model.**

*Although we only tested one microphysics scheme, we demonstrated that a modern, complex parameterization does not imply accurate representation of* **all** *cloud microphysical properties and suggest that calculations of the radiative forcing of these phenomena* **may be biased under polluted conditions like those found during the Amazon biomass burning season.**

**Evaluation of the parameterization of Freud et al. (2011) proved to be successful in deriving $N_a$ from cloud-side remote sensing data collected by the specMACS instrument. We note a high sensitivity of the method at low $N_a$ and its dependence on an average mixing factor. We were able to gain these insights by applying a previously developed parameterization in a new context. Our study demonstrates that, despite some inherent challenges, existing techniques can be applied for model-measurement** *comparisons to* **improve our understanding** *of model biases."*

**Minor comments**

The Reid et al, 1999 paper is highly relevant to this study and should not be brought up for the first time in the conclusion. The authors should discuss the main findings of the paper in the introduction, and put their results more fully in the context of Reid's work in the discussion.

We agree that Reid et al. (1999) and their findings should be mentioned earlier, and have therefore added a paragraph to the introduction:

*"[...] cloud droplet effective radius ($r_{eff}$) vertical profiles, since $r_{eff}$ profiles represent the micro-physical development of a cloud and can be derived from in situ and remote sensing observations.*

*Reid et al. (1999) similarly investigated the effects of biomass burning in Brazil. In their simulations, they found no further changes in $r_{eff}$ from additional biomass burning aerosol when regional background accumulation-mode aerosol concentration reached 3000-4000 $cm^{-3}$. $r_{eff}$ was then merely a function of the liquid water content. They also showed that $r_{eff}$ for clouds affected by biomass burning smoke are considerable smaller than those of clouds in more pristine environments like a marine boundary layer.*
*Though $r_{eff}$ profiles describe the vertical evolution[...]"*

L37 "nucleii"->"nuclei"

Corrected.

L371 "measured my"->"measured by"

Corrected.

**The challenge of simulating the sensitivity of the Amazonian clouds microstructure to cloud condensation nuclei number concentrations**

Pascal Polonik[1,*], Christoph Knote[1], Tobias Zinner[1], Florian Ewald[2], Tobias Kölling[1], Bernhard Mayer[1], Meinrat O. Andreae[3,4], Tina Jurkat-Witschas[4], Thomas Klimach[4], Christoph Mahnke[5,6], Sergej Molleker[5], Christopher Pöhlker[4], Mira L. Pöhlker[4], Ulrich Pöschl[4], Daniel Rosenfeld[7], Christiane Voigt[2,6], Ralf Weigel[6], and Manfred Wendisch[8]

[1]Meteorologisches Institut, Ludwig Maximilians-Universität München, Munich, Germany
[2]Institut für Physik der Atmosphäre, Deutsches Zentrum für Luft- und Raumfahrt (DLR), Oberpfaffenhofen, Germany
[3]Scripps Institution of Oceanography, University of California at San Diego, La Jolla, California, USA
[4]Multiphase Chemistry and Biogeochemistry Departments, Max Planck Institute for Chemistry, Mainz, Germany
[5]Particle Chemistry Department, Max Planck Institute for Chemistry, Mainz, Germany
[6]Institut für Physik der Atmosphäre, Johannes Gutenberg-Universität, Mainz, Germany
[7]Institute of Earth Sciences, Hebrew University of Jerusalem, Jerusalem, Israel
[8]Leipziger Institut für Meteorologie, Universität Leipzig, Leipzig, Germany
[*]*now at:* Scripps Institution of Oceanography, University of California at San Diego, La Jolla, California, USA

**Correspondence:** Christoph Knote (christoph.knote@physik.uni-muenchen.de)

**Abstract.** The realistic representation of  aerosol-cloud interactions is of primary importance for accurate climate model projections. The investigation of these interactions in strongly contrasting clean and polluted atmospheric conditions in the Amazon region has been one of the motivations for several field campaigns, including the airborne Aerosol, Cloud, Precipitation, and Radiation Interactions and DynamIcs of CONvective cloud systems - Cloud Processes of the Main Precipitation Systems in Brazil: A Contribution to Cloud Resolving Modeling and to the GPM (Global Precipitation Measurement) (ACRIDICON-CHUVA) campaign based in Manaus, Brazil in September 2014. In this work we combine in situ and remotely sensed aerosol, cloud, and atmospheric radiation data collected during ACRIDICON-CHUVA with regional, online-coupled chemistry-transport simulations to evaluate the model's ability to represent the indirect effects of biomass burning aerosol on cloud microphysical and optical properties (droplet number concentration and effective radius).

We found agreement between modeled and observed median cloud droplet number concentrations (CDNC) for low values of CDNC, i.e., low levels of pollution. In general, a linear relationship between  modelled and observed CDNC with a slope of  0.3 was found, which  implies a systematic underestimation of modeled CDNC as compared to measurements. Variability in cloud condensation nuclei (CCN) number concentrations was also underestimated and cloud droplet effective radii ($r_{\text{eff}}$)  were overestimated by the model.

Modeled effective radius profiles began to saturate around 500 CCN $\text{per cm}^3$ at cloud base, indicating an upper limit for the model sensitivity well below CCN concentrations reached during the burning season in the Amazon Basin.  Additional CCN emitted from local fires did not cause a notable change in modelled cloud

 droplet effective radii. Finally, we also evaluate a parameterization of CDNC at cloud base using more readily available cloud microphysical properties. , showing that we are able to derive CDNC at cloud base from cloud-side remote sensing observations.

*Copyright statement.* TEXT

**1 Introduction**

[revised manuscript text omitted]
)  have demonstrated substantial Twomey effects at much higher aerosol loadings. Additionally, the relation between modelled and observed CDNC is linear and has a slope of 0.3, indicating a considerable underestimation of cloud droplet number concentrations by the model. Although we only tested one microphysics scheme, we demonstrated that

365   a modern, complex parameterization does not imply accurate representation of all cloud microphysical properties and suggest that calculations of the radiative forcing of these phenomena  may be biased under polluted conditions like those found during the Amazon biomass burning season.

Evaluation of the parameterization of Freud et al. (2011) proved to be successful in deriving $N_a$ from cloud-side remote sensing data collected by the specMACS instrument. We note a high sensitivity of the method at low $N_a$ and its dependence

370   on an average mixing factor. We were able to gain these insights by applying a previously developed parameterization in a new context. Our study demonstrates that, despite some inherent challenges, existing techniques can be applied for model-measurement comparisons to  improve our understanding of model biases.

[revised manuscript text omitted]